# Pre-Training Graph Contrastive Masked Autoencoders are Strong Distillers for EEG

## Abstract

Effectively utilizing extensive unlabeled high-density EEG data to improve performance in scenarios with limited labeled low-density EEG data presents a significant challenge. In this paper, we address this by framing it as a graph transfer learning and knowledge distillation problem. We propose a Unified Pre-trained Graph Contrastive Masked Autoencoder Distiller, named EEG-DisGCMAE, to bridge the gap between unlabeled/labeled and high/low-density EEG data. To fully leverage the abundant unlabeled EEG data, we introduce a novel unified graph self-supervised pre-training paradigm, which seamlessly integrates Graph Contrastive Pre-training and Graph Masked Autoencoder Pre-training. This approach synergistically combines contrastive and generative pre-training techniques by reconstructing contrastive samples and contrasting the reconstructions. For knowledge distillation from high-density to low-density EEG data, we propose a Graph Topology Distillation loss function, allowing a lightweight student model trained on low-density data to learn from a teacher model trained on high-density data, effectively handling missing electrodes through contrastive distillation. To integrate transfer learning and distillation, we jointly pre-train the teacher and student models by contrasting their queries and keys during pre-training, enabling robust distillers for downstream tasks. We demonstrate the effectiveness of our method on four classification tasks across two clinical EEG datasets with abundant unlabeled data and limited labeled data. The experimental results show that our approach significantly outperforms contemporary methods in both efficiency and accuracy.

## 1 Introduction

Electroencephalography (EEG) is a pivotal tool for elucidating neural dysfunctions, making it indispensable for the clinical diagnosis of brain disorders (Sanei & Chambers, 2013). Manual analysis of resting-state EEG (rs-EEG) signals often suffers from low accuracy due to their inherent complexity. In contrast, computer-aided diagnostic methods offer substantial improvements in diagnostic performance. Traditional methods typically involve extracting temporal and spatial features from EEG signals and applying machine learning techniques to develop effective classifiers (Trivedi et al., 2016). Recent advances have seen deep graph learning revolutionize EEG signal analysis. Instead of treating EEG data as conventional numerical inputs, researchers now represent it as non-Euclidean graph data. Graph Neural Networks (GNNs) (Kipf & Welling, 2016) are employed to capture the intricate features and topological structures inherent in these graphs. This innovative approach has markedly enhanced the accuracy and reliability of EEG-based diagnostics, showcasing the potential of GNNs in advancing applications (Song et al., 2018).

Despite these advancements, several critical issues remain unresolved. Firstly, acquiring a substantial amount of accurately labeled clinical rs-EEG data for supervised training on a specific task is challenging due to the complexities involved in data collection (Siuly et al., 2016). Models trained on these limited labeled datasets often exhibit poor accuracy and generalization (Lashgari et al., 2020). Thus, a significant but underexplored research problem is how to effectively utilize this vast amount of unlabeled data to enhance model performance and robustness (Tang et al., 2021). Secondly, the performance of EEG devices varies markedly with the precision of the data they capture. High-density (HD) EEG devices, with their extensive array of electrodes, record high-resolution brain signals, greatly improving the accuracy of diagnostic tasks (Stoyell et al., 2021). However,

these devices are often prohibitively expensive and cumbersome, limiting their practical deployment. Conversely, low-density (LD) EEG devices, which are more affordable and easier to deploy (Justesen et al., 2019), capture lower-resolution signals, thus reducing diagnostic accuracy (Cataldo et al., 2022). Addressing how to leverage rich information from HD EEG to enhance diagnostic performance with LD EEG, which is more portable, is crucial for making LD EEG-based diagnostics more accessible and practical (Kuang et al., 2021).

In this paper, we address these challenges through a series of innovative methods. We construct graphs from EEG data and apply GNNs to extract topological features and train the model effectively. To leverage unlabeled EEG data to enhance performance on limited labeled data, we frame this as a Graph Transfer Learning (GTL) problem. We propose a graph self-supervised pre-training (PT) approach (Xie et al., 2022) on a large volume of heterogeneous unlabeled EEG graphs. This pre-trained model is subsequently fine-tuned (FT) on the scarce labeled data, allowing knowledge acquired from the extensive unlabeled dataset to improve performance on the labeled data. We introduce a novel unified graph self-supervised pre-training paradigm, GCMAE-PT, which combines Graph Contrastive Pre-training (GCL-PT) (Qiu et al., 2020) with Graph Masked Autoencoder Pre-training (GMAE-PT) (Hou et al., 2022). This approach integrates contrastive and generative pre-training by reconstructing contrastive samples and contrasting the reconstructed samples, enabling them to jointly supervise and optimize each other, thereby enhancing overall model performance. To improve model performance with HD EEG data when training on LD EEG data, we address this as a Graph Knowledge Distillation (GKD) problem (Yang et al., 2020) and design a Graph Topology Distillation (GTD) loss function. This allows a student model trained on LD EEG to learn from a teacher model with HD EEG by accounting for missing electrodes through contrastive distillation, while simultaneously compressing model parameters. Moreover, to ensure that models pre-trained with GTL excel as distillers in downstream GKD tasks, we integrate GTL and GKD by contrasting the queries and keys of the teacher and student models during the GTL pre-training process. This integration demonstrates that our unified pre-trained graph contrastive masked autoencoders serve as effective distillers, providing a robust solution for EEG analysis.

## 2 RELATED WORKS

### 2.1 GRAPH NEURAL NETWORKS FOR EEG MODELING

Recent advancements in Graph Neural Networks (GNNs) have demonstrated their potential in enhancing the modeling and interpretation of EEG data. Notably, the Dynamical Graph Convolutional Neural Network (DGCNN) (Song et al., 2018) was introduced to improve emotion recognition by dynamically learning the interrelationships among EEG channels. Similarly, the Regularized Graph Neural Network (RGNN) (Zhong et al., 2020) applied a regularization strategy to advance emotion recognition from EEG data. Liu et al. (Liu et al., 2023) tackled a similar problem by developing a novel method for emotion recognition from few-channel EEG signals, integrating deep feature aggregation with transfer learning. For medical EEG field, Tang et al. (Tang et al., 2021) employed self-supervised GNNs to advance seizure detection and classification, achieving significant improvements in identifying rare seizure types.

### 2.2 SELF-SUPERVISED GRAPH PRE-TRAINING

Self-supervised learning (SSL) pre-training (Zhang et al., 2022a) has proven effective in harnessing extensive unlabeled datasets. Two predominant SSL methods are contrastive learning-based (CL-PT) pre-training, originating from computer vision, and generative-based masked autoencoders (MAE-PT) pre-training, adapted from natural language processing (NLP). These pre-training techniques have been extended to graph models. For instance, GCC (Qiu et al., 2020) and GraphCL (You et al., 2020) were among the pioneers in applying contrastive learning to graphs by leveraging graph augmentation to generate sample pairs and construct contrastive losses. Concurrently, GMAE (Hou et al., 2022) and GPT-GNN (Hu et al., 2020) adapted the generative masked pre-training approach from NLP (Devlin et al., 2018) to graphs. These methods involve masking nodes and edges, followed by reconstruction, enabling graphs to capture and refine local topological features.

## 2.3 GRAPH KNOWLEDGE DISTILLATION

Graph Knowledge Distillation focuses on transferring knowledge from a complex, large-scale model (teacher) to a more streamlined and efficient model (student), thus preserving performance while reducing computational demands. G-CRD (Joshi et al., 2022) introduced a distillation loss function for GNN-to-GNN transfer, employing a contrastive learning strategy to enhance similarity among nodes of the same class and increase separation between different classes. MSKD (Zhang et al., 2022b) proposed a multi-teacher distillation approach, integrating various teacher GNN models of different scales into a single student GNN model. Approaches such as Graph-MLP (Hu et al., 2021), and VQGraph (Yang et al., 2024) focused on transferring knowledge from structure-aware teacher GNNs to structure-agnostic student MLPs.

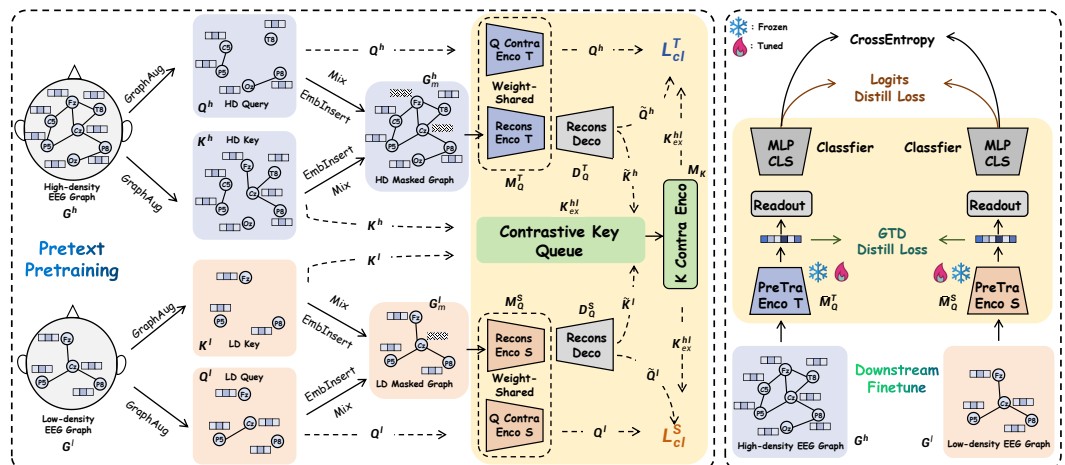

Figure 1: The proposed EEG-DisGCMAE framework consists of two main stages: a pretext pre-training (PT) stage and a downstream fine-tuning (FT) stage. Note that we can perform two types of fine-tuning: 'Tuned' refers to fine-tuning all the parameters of the model, while 'Frozen' means freezing most layers of the model and only fine-tuning the parameters of the top-level layers. Note that the encoders of our model can be adopted *Graph Transformer* or *vanilla GCNs*.

## 3 METHODOLOGY

### 3.1 EEG GRAPH CONSTRUCTION

An EEG graph can be formally represented as $\mathcal{G} = (\mathcal{V}, \mathcal{A})$, where $\mathcal{V} = \{v_1, \ldots, v_n\}$ denotes the set of nodes in the graph $\mathcal{G}$. The matrix $\mathcal{X} \in \mathbb{R}^{n \times d}$ represents the node features, with $n$ indicating the number of nodes (or electrodes) and $d$ specifying the dimensionality of the feature vector associated with each node. The adjacency matrix of the EEG graph $\mathcal{G}$ is denoted by $\mathcal{A} \in \mathbb{R}^{n \times n}$. The EEG graph $\mathcal{G}$ is derived from the original EEG time series signals $\mathcal{S} = \{s_1, \ldots, s_n\} \in \mathbb{R}^{n \times t}$ recorded by EEG caps, where $n$ represents the number of channels or electrodes, and $t$ denotes the length of the time series for each channel.

$$\mathcal{X} = PSD(Filter(\mathcal{S})) \qquad \mathcal{A} = Corr(\mathcal{X}) \qquad (1)$$

To convert resting-state EEG (rs-EEG) time series into graph representations, we first apply band-pass filtering $Filter(\cdot)$ to extract EEG signals within the following frequency bands: $\theta$ (4-8 Hz), $\alpha$ (8-14 Hz), $\beta$ (14-30 Hz), and $\gamma$ (30-50 Hz). Subsequently, we compute the power spectral density (PSD) features $PSD(\cdot)$ for each band, selecting the $\alpha$ band for this study. These PSD features are utilized as node features for the EEG graph. The Pearson correlation $Corr(\cdot)$ is then computed between nodes to construct the adjacency matrix $\mathcal{A}$, which represents the edge connectivity.

### 3.2 UNIFIED GRAPH PRE-TRAINING FOR DISTILLATION

To fully leverage the extensive amount of unlabeled EEG data, we propose a graph self-supervised pre-training approach to pre-train EEG models from a graph-based perspective. Our *motivation*

stems from the observation that prior research has predominantly focused on either contrastive-based or generative-based pre-training methods for EEG time series, with limited studies addressing these techniques within the context of EEG graph models. To address this gap, we introduce a unified graph self-supervised pre-training paradigm, termed GCMAE-PT, based on the following assumptions:

***Assumption 1***: *(Combining GCL and GMAE for Enhanced Distillation) Hybridizing contrastive-based and generative-based pre-training by simultaneously reconstructing contrastive pairs and contrasting the reconstructed samples provides a more robust distiller, rather than applying these methods separately or in sequence.*

***Assumption 2***: *(Joint Pre-Training of Teacher and Student Models) Both the teacher and student models benefit from joint pre-training through the contrasting of each other's positive and negative pairs, leading to improved distillation performance.*

Consider two types of EEG graph inputs: the high-density EEG graph $\mathcal{G}^h = (\mathcal{V}^h, \mathcal{A}^h) \in \mathbb{R}^{m \times d}$ with nodes $\mathcal{V}^h = \{v_1^h, \ldots, v_m^h\}$ and the low-density EEG graph $\mathcal{G}^l = (\mathcal{V}^l, \mathcal{A}^l) \in \mathbb{R}^{n \times d}$ with nodes $\mathcal{V}^l = \{v_1^l, \ldots, v_n^l\}$, where $m$ and $n$ represent the number of nodes (or electrodes) in $\mathcal{G}^h$ and $\mathcal{G}^l$, respectively, and $m \geq n$. Note that $\mathcal{G}^l$ can be regarded as a subgraph of $\mathcal{G}^h$. Additionally, two graph encoders are employed: a teacher graph encoder $\mathcal{M}_Q^T$ (where Q denotes queries) with extensive parameters and robust feature extraction capabilities, and a lightweight student graph encoder $\mathcal{M}_Q^S$ with fewer parameters and comparatively lower learning capacity. It is noteworthy that $\mathcal{A}^h$ and $\mathcal{A}^l$ can be dynamically learned and adjusted throughout the training process. The teacher and student models are adaptable to different types of GNNs, such as transductive spectral-based traditional GCNs (like DGCNN(Song et al., 2018)) or spatial-based graph transformers (Yun et al., 2019).

Since $\mathcal{V}^l$ is derived from $\mathcal{V}^h$, that is $\mathcal{V}^l \subseteq \mathcal{V}^h$, we partition the complete node set $\mathcal{V}^h$ (the Complete/HD Set) in $\mathcal{G}^h$ into *the Deleted Set* $\mathcal{V}^d = v_1^d, \ldots, v_{m-n}^d$ and *the Remaining/LD Set* $\mathcal{V}^l$. The set $\mathcal{V}^d$ comprises $(m - n)$ nodes present in $\mathcal{G}^h$ but absent from $\mathcal{G}^l$, representing the removed electrodes/nodes. Conversely, $\mathcal{V}^l$ includes the $n$ nodes retained in $\mathcal{G}^l$. The relationships among these sets can be expressed as $\mathcal{V}^l = \mathcal{V}^h - \mathcal{V}^d$, where $\mathcal{V}^l \subseteq \mathcal{V}^h$, $\mathcal{V}^d \subseteq \mathcal{V}^h$, $\mathcal{V}^d \cap \mathcal{V}^l = \varnothing$, and $\mathcal{V}^d \cup \mathcal{V}^l = \mathcal{V}^h$. Thus, the complete set $\mathcal{V}^h$ is composed of the deleted set $\mathcal{V}^d$ and the remaining LD set $\mathcal{V}^l$.

As illustrated in Fig. 1, to construct the contrastive-based pre-training paradigm, graph augmentation techniques *Aug(.)* (You et al., 2020) are initially applied to $\mathcal{G}^h$ and $\mathcal{G}^l$ by randomly dropping nodes (i.e., count = c) and removing edges. This process yields *Query graphs* ($\mathcal{Q}^h = q_1^h, \ldots, q_{m-c}^h \in \mathbb{R}^{(m-c) \times d}$, $\mathcal{Q}^l = q_1^l, \ldots, q_{n-c}^l \in \mathbb{R}^{(n-c) \times d}$) and *Key graphs* ($\mathcal{K}^h = k_1^h, \ldots, k_{m-c}^h \in \mathbb{R}^{(m-c) \times d}$, $\mathcal{K}^l = k_1^l, \ldots, k_{n-c}^l \in \mathbb{R}^{(n-c) \times d}$) for both $\mathcal{G}^h$ and $\mathcal{G}^l$. This can be formulated as $(\mathcal{Q}^h, \mathcal{K}^h) = Aug(\mathcal{G}^h)$ and $(\mathcal{Q}^l, \mathcal{K}^l) = Aug(\mathcal{G}^l)$. Consequently, the total augmented graphs for $\mathcal{G}^h$ and $\mathcal{G}^l$ are denoted as $\widehat{\mathcal{G}}^h$ and $\widehat{\mathcal{G}}^l$. Note that $\widehat{\mathcal{G}}^h = Mix(\mathcal{Q}^h, \mathcal{K}^h) = \mathcal{Q}^h, \mathcal{K}^h$ and $\widehat{\mathcal{G}}^l = Mix(\mathcal{Q}^l, \mathcal{K}^l) = \mathcal{Q}^l, \mathcal{K}^l$, where the function *Mix(.)* represents the integration of two graph sets.

To achieve the goal of *reconstructing the contrastive pairs* as outlined in *Assumption 1*, the *Masked Graphs* $\mathcal{G}_m^h$ and $\mathcal{G}_m^l$ for GMAE-PT are constructed from the mixed contrastive augmented samples $\widehat{\mathcal{G}}^h$ and $\widehat{\mathcal{G}}^l$ by substituting the dropped nodes $v_i^d$ from $\mathcal{V}^d$ with learnable embeddings $e_i \in \mathbb{R}^{1 \times d}$. Subsequently, both the teacher and student encoders are employed to encode the masked graphs $\mathcal{G}_m^h$ and $\mathcal{G}_m^l$ into the graph embeddings. To accomplish GMAE-PT, graph decoders $\mathcal{D}_Q^T$ and $\mathcal{D}_Q^S$ for both teacher and student encoders are utilized to reconstruct the masked graph embeddings into the original inputs $\mathcal{G}^h$ and $\mathcal{G}^l$ by applying the *MSE Loss* as the reconstruction loss $\mathcal{L}_{Rec}$ on both the reconstructed node features $\tilde{\mathcal{X}}$ and graph structures $\tilde{A} = \tilde{\mathcal{X}} \cdot \tilde{\mathcal{X}}^{tr}$ (Yang et al., 2024).

$$\mathcal{L}_{Rec} = \left\| \mathcal{X} - \tilde{\mathcal{X}} \right\|_2^2 + \left\| A - \tilde{\mathcal{X}} \cdot \tilde{\mathcal{X}}^{tr} \right\|_2^2 \tag{2}$$

where $\tilde{\mathcal{X}}^{tr}$ means the transpose of $\tilde{\mathcal{X}}$. Then the reconstructed HD and LD query ($\tilde{\mathcal{Q}}^h$, $\tilde{\mathcal{Q}}^l$) and key ($\tilde{\mathcal{K}}^h$, $\tilde{\mathcal{K}}^l$) graphs are split out from the reconstructed $\tilde{\mathcal{G}}^h$ and $\tilde{\mathcal{G}}^l$.

To achieving the goal of *contrasting the reconstructed samples* in *Assumption 1*, the reconstructed HD and LD query ($\tilde{\mathcal{Q}}^h = \{\tilde{q}_1^h, \ldots, \tilde{q}_m^h\}$, $\tilde{\mathcal{Q}}^l = \{\tilde{q}_1^l, \ldots, \tilde{q}_n^l\}$) and key ($\tilde{\mathcal{K}}^h = \{\tilde{k}_1^h, \ldots, \tilde{k}_m^h\}$, $\tilde{\mathcal{K}}^l = \{\tilde{k}_1^l, \ldots, \tilde{k}_n^l\}$) graphs are mixed with the original contrastive samples generated by augmentation as additional contrastive HD and LD query and key samples to form the *extended* contrastive HD and

LD query $(\mathcal{Q}_{ex}^h, \mathcal{Q}_{ex}^l)$ and key $(\mathcal{K}_{ex}^h, \mathcal{K}_{ex}^l)$ samples.

$$\mathcal{Q}_{ex}^h = \{\mathcal{Q}^h, \tilde{\mathcal{Q}}^h\} \qquad \mathcal{Q}_{ex}^l = \{\mathcal{Q}^l, \tilde{\mathcal{Q}}^l\} \qquad \mathcal{K}_{ex}^h = \{\mathcal{K}^h, \tilde{\mathcal{K}}^h\} \qquad \mathcal{K}_{ex}^l = \{\mathcal{K}^l, \tilde{\mathcal{K}}^l\} \qquad (3)$$

To achieve the goal of *joint the teacher and student pre-training via contrasting the reconstructed samples* in *Assumption 2*, the extended key samples of both HD and LD graphs $\mathcal{K}_{ex}^h$ and $\mathcal{K}_{ex}^l$ are mixed to form a larger *Key Samples Pool* $\mathcal{K}_{ex}^{hl}$.

$$\mathcal{K}_{ex}^{hl} = \{\mathcal{K}^h, \tilde{\mathcal{K}}^h, \mathcal{K}^l, \tilde{\mathcal{K}}^l\} = KQ(\{\mathcal{K}_{ex}^{hl+}, \mathcal{K}_{ex}^{hl-}\}) \qquad (4)$$

Following (He et al., 2020; Qiu et al., 2020), we adopt a *Key Queue*, denoted as *KQ(.)*, to store a large number of mixed extended *key samples pool* $\mathcal{K}_{ex}^{hl}$ and *key encoders* for both teacher and student to convert $\mathcal{K}_{ex}^{hl}$ to be key embeddings for jointly pre-training the teacher and student encoders via a joint contrastive loss function (Qiu et al., 2020) as follows:

$$\mathcal{L}_{cl}^T = -\log\left(\frac{\exp(\mathcal{Q}_{ex}^h \cdot \mathcal{K}_{ex}^{hl+}/\tau)}{\sum_{i=0}^K \exp(\mathcal{Q}_{ex}^h \cdot \mathcal{K}_{ex}^{hl-}/\tau)}\right) \qquad \mathcal{L}_{cl}^S = -\log\left(\frac{\exp(\mathcal{Q}_{ex}^l \cdot \mathcal{K}_{ex}^{hl+}/\tau)}{\sum_{i=0}^K \exp(\mathcal{Q}_{ex}^l \cdot \mathcal{K}_{ex}^{hl-}/\tau)}\right) \qquad (5)$$

where $\tau$ represents the temperature coefficient.

Then, we simultaneously contrast the extended HD queries $\mathcal{Q}_{ex}^h$ and LD queries $\mathcal{Q}_{ex}^l$ with all the mixed extended keys $\mathcal{K}_{ex}^{hl}$ in the queue, which consists of both HD and LD keys with the corresponding HD and LD reconstructed keys, to construct the positive and negative pairs with their corresponding positive and negative keys $\{\mathcal{K}_{ex}^{hl+}, \mathcal{K}_{ex}^{hl-}\}$ in the queue, computing contrastive loss to jointly optimize the query and key encoders of both teacher and student models. Therefore, the joint contrastive loss function for GCL-PT $\mathcal{L}_{cl}^{Joint}$ is composed of the teacher contrastive loss $\mathcal{L}_{cl}^T$ and the student contrastive loss $\mathcal{L}_{cl}^S$. And the joint reconstruction loss function $\mathcal{L}_{Rec}^{Joint}$ for GMAE-PT consists of the teacher reconstruction loss $\mathcal{L}_{Rec}^T$ and the student reconstruction loss $\mathcal{L}_{Rec}^S$ as follows:

$$\mathcal{L}_{cl}^{Joint} = \mathcal{L}_{cl}^T + \mathcal{L}_{cl}^S \qquad \mathcal{L}_{Rec}^{Joint} = \mathcal{L}_{Rec}^T + \mathcal{L}_{Rec}^S \qquad (6)$$

The overall loss $\mathcal{L}_{Pretrain}$ for both the teahcer and student encoders pre-training is composed of the contrastive-based loss $\mathcal{L}_{cl}^{Joint}$ and the generative-based loss $\mathcal{L}_{Rec}^{Joint}$.

$$\mathcal{L}_{Pretrain} = \mathcal{L}_{cl}^{Joint} + \mathcal{L}_{Rec}^{Joint} + \mathcal{L}_{Dis}^{GTD} \qquad (7)$$

Note that the $\mathcal{L}_{Dis}^{GTD}$ is proposed to distill the structure information, which will be demonstrated in details in the next section.

### 3.3 GRAPH TOPOLOGY DISTILLATION FOR HD-LD EEG

In the downstream stage, the pre-trained models $\bar{\mathcal{M}}_Q^T$ and $\bar{\mathcal{M}}_Q^S$ are fine-tuned for specific classification tasks using limited labeled EEG data from $\mathcal{G}^h$ and $\mathcal{G}^l$. We employ the Cross-Entropy loss $\mathcal{L}_{CE}$ for classification. To transfer logit-based knowledge, we adopt the classic logit distillation loss $\mathcal{L}_{Dis}^{logits}$ (Hinton et al., 2015), using KL divergence *KL(.)* to align the predicted logit distributions, allowing $\bar{\mathcal{M}}_Q^S$ to mimic the logits of $\bar{\mathcal{M}}_Q^T$. Moreover, since $\mathcal{G}^h$ contains more nodes than $\mathcal{G}^l$, the topological information $\mathcal{A}^h$ learned by $\bar{\mathcal{M}}_Q^T$ from the high-density graph $\mathcal{G}^h$ is more precise and discriminative than $\mathcal{A}^l$, learned by $\bar{\mathcal{M}}_Q^S$ from the low-density graph $\mathcal{G}^l$. These topological features capture the spatial connectivity of EEG electrodes, which is crucial for task performance. Thus, distilling the topological knowledge from $\bar{\mathcal{M}}_Q^T$ into $\bar{\mathcal{M}}_Q^S$ is essential to boost the performance of $\bar{\mathcal{M}}_Q^S$. To address this, we propose the Graph Topology Distillation loss $\mathcal{L}_{Dis}^{GTD}$.

To quantify the similarity between node features $\mathcal{X}_i$ of node $v_i$ and $\mathcal{X}_j$ of node $v_j$ in the graph, we employ a similarity kernel function (Joshi et al., 2022). This function computes the similarity $\mathcal{Z}_{ij}$ for both $\mathcal{G}^h$ and $\mathcal{G}^l$. Specifically, we adopt the *Linear Kernel* as the node similarity function $\mathcal{F}(.)$, defined as follows:

$$\mathcal{Z}_{ij}^h = \mathcal{F}(\mathcal{X}_i^h, \mathcal{X}_j^h) = \mathcal{X}_i^h \cdot \mathcal{X}_j^h \qquad \mathcal{Z}_{ij}^l = \mathcal{F}(\mathcal{X}_i^l, \mathcal{X}_j^l) = \mathcal{X}_i^l \cdot \mathcal{X}_j^l \qquad (8)$$

Note that $\{v_i, v_j\} \in (\mathcal{V}^h \cap \mathcal{V}^l)$ and $\{v_i, v_j\} \notin \mathcal{V}^d$.

Guided by positive and negative pairs in $\mathcal{A}^h$ and the influence of the $\mathcal{L}_{Dis}^{GTD}$ loss, we aim to pull similar positive node pairs $\mathcal{P}_{ij}^+$ closer and push dissimilar negative node pairs $\mathcal{P}_{ij}^-$ farther apart in $\mathcal{A}^l$. This process first requires defining and selecting $\mathcal{P}_{ij}^+$ and $\mathcal{P}_{ij}^-$ for both $\mathcal{G}^h$ and $\mathcal{G}^l$. As described earlier, three node sets are involved: the complete/HD set, the deleted/removed set, and the remaining/LD set. Since $\mathcal{G}^l$ is formed by removing certain electrodes/channels/nodes $\mathcal{V}^d$ from $\mathcal{V}^h$, the removed electrodes $\mathcal{V}^d$ significantly influence the topological structure of $\mathcal{G}^l$. We define the positive and negative contrastive pairs as follows:

$$\mathcal{P}_{ij}^+ = \mathbb{I}\left(\mathcal{A}_{ij}^h > 0 \text{ or } \exists k \in \mathcal{V}^d \text{ s.t. } \mathcal{A}_{ik}^h > 0 \text{ and } \mathcal{A}_{kj}^h > 0\right) \tag{9}$$

$$\mathcal{P}_{ij}^- = \mathbb{I}\left(\mathcal{A}_{ij}^l > 0 \text{ and } \left(\mathcal{A}_{ij}^h = 0 \text{ and } \forall k \in \mathcal{V}^d, \mathcal{A}_{ik}^h = 0 \text{ or } \mathcal{A}_{kj}^h = 0\right)\right) \tag{10}$$

where $\mathbb{I}$ represents the Indicator function.

**Positive and Negative Nodes Selection:** As described in the equations above, in $\mathcal{G}^l$, if two nodes $v_i$ and $v_j$ are either directly connected (1-hop) or indirectly connected (2-hop) through a removed node $v_k$ in $\mathcal{V}^d$, acting as a mediator in the graph embedding of $\mathcal{G}^h$ learned by $\bar{\mathcal{M}}_Q^T$, these node pairs $v_{ij}$ are treated as positive contrastive pairs. Conversely, if node pairs $v_{ij}$ are connected in the embedding learned by $\bar{\mathcal{M}}_Q^S$ but are neither directly nor indirectly connected in the embedding learned by $\bar{\mathcal{M}}_Q^T$, they are treated as negative contrastive pairs. Once the positive and negative pairs $\mathcal{P}_{ij}^+$ and $\mathcal{P}_{ij}^-$ are identified, we apply KL divergence $KL(.)$ as the distillation function. In the numerator, it is used to align the kernel feature distributions learned by $\bar{\mathcal{M}}_Q^T$ and $\bar{\mathcal{M}}_Q^S$ for positive pairs, encouraging $\bar{\mathcal{M}}_Q^S$ to replicate the topological distribution of $\bar{\mathcal{M}}_Q^T$ and increase the similarity of positive pairs in the embeddings learned by $\bar{\mathcal{M}}_Q^S$. In the denominator, KL divergence is also employed to adjust the erroneous topological distribution learned from negative pairs by $\bar{\mathcal{M}}_Q^S$.

$$\mathcal{L}_{Pos} = \sum_{(i,j) \in \mathcal{P}^+} KL\left(softmax(\mathcal{Z}_{ij}^l) \| softmax(\mathcal{Z}_{ij}^h)\right)$$
$$\mathcal{L}_{Neg} = \sum_{(i,j) \in \mathcal{P}^-} KL\left(softmax(\mathcal{Z}_{ij}^l) \| softmax(\mathcal{Z}_{ij}^h)\right) \tag{11}$$

The final GTD loss function $\mathcal{L}_{Dis}^{GTD}$ in the contrastive format is as follows:

$$\mathcal{L}_{Dis}^{GTD} = \frac{\mathcal{L}_{Pos}/\mathcal{C}_{Pos}}{\mathcal{L}_{Neg}/\mathcal{C}_{Neg} + \epsilon} \tag{12}$$

where $\mathcal{C}_{Pos}$ and $\mathcal{C}_{Neg}$ are the counts of $\mathcal{P}_{ij}^+$ and $\mathcal{P}_{ij}^-$. $\epsilon$ is a constant to avoid division by zero errors.

Finally, we integrate our $\mathcal{L}_{Dis}^{GTD}$ with $\mathcal{L}_{Dis}^{logits}$ and $\mathcal{L}_{CE}$ to form the total *Fine-tune* loss $\mathcal{L}_{Finetune}$:

$$\mathcal{L}_{Finetune} = \mathcal{L}_{CE} + \mathcal{L}_{Dis}^{Logits} + \mathcal{L}_{Dis}^{GTD} \tag{13}$$

Note that $\mathcal{L}_{Dis}^{GTD}$ is also be adopted in the pre-training stage.

### 3.4 SPECIAL CASE FOR THE PROPOSED GTD LOSS

The GTD loss is primarily designed to distill topological knowledge from $\mathcal{G}^h$ to $\mathcal{G}^l$. However, there is a special case known as H2H distillation, where $\mathcal{G}^l$ and $\mathcal{G}^h$ have the same number of nodes, meaning $\mathcal{V}^l = \mathcal{V}^h$ and $\mathcal{V}^d = \varnothing$. In this scenario, no nodes are removed, and only the connections in $\mathcal{A}^l$ and $\mathcal{A}^h$ may differ. With slight modifications, our loss function can also be applied to this special case. The modified GTD loss for the H2H distillation scenario is given as follows:

$$\mathcal{P}_{ij}^+ = \mathbb{I}\left(\mathcal{A}_{ij}^h > 0\right) \qquad \mathcal{P}_{ij}^- = \mathbb{I}\left(\mathcal{A}_{ij}^l > 0 \text{ and } \mathcal{A}_{ij}^h = 0\right) \tag{14}$$

In this special case, GTD loss does not consider $\mathcal{V}^d$. The learning objective becomes utilizing the learned $\mathcal{A}^h$ learned from $\bar{\mathcal{M}}_Q^T$ to correct incorrectly edges in $\mathcal{A}^h$ learned from $\bar{\mathcal{M}}_Q^S$, thereby making $\mathcal{A}^l$ as close to $\mathcal{A}^h$ as possible.

## 4 Experiments

### 4.1 EEG Datasets and Downstream Tasks

We evaluated our EEG-DisGCMAE framework on two clinical datasets with rs-EEG time series: the Establishing Moderators and Biosignatures of Antidepressant Response in Clinical Care (EM-BARC) (Trivedi et al., 2016) and the Healthy Brain Network (HBN) (Alexander et al., 2017). The EMBARC dataset comprises EEG data from 308 eye-open and 308 eye-closed samples, while the HBN dataset includes 1,594 eye-open and 1,745 eye-closed samples. Detailed dataset preprocessing information is provided in the appendices. For EMBARC, we performed binary classification tasks: sex classification in Major Depressive Disorder (MDD) patients (Male vs Female) and depression severity classification based on the Hamilton Depression Rating Scale ($HAMD_{17}$) (Williams, 1988) (Mild vs Severe Depression) (Boessen et al., 2013). For HBN, we conducted binary classifications for MDD (Healthy vs MDD) and Autism Spectrum Disorder (ASD) (Healthy vs ASD). Additional details can be found in the appendices. We tested three EEG electrode density levels: high-density (HD), medium-density (MD), and low-density (LD). In EMBARC, these densities correspond to the 10-20 EEG system electrode distributions of 64 (HD), 32 (MD), and 16 (LD) electrodes, respectively. For HBN, the densities correspond to 128 (HD), 64 (MD), and 32 (LD) electrodes.

### 4.2 Comparative Experiment Analysis

We compared the proposed EEG-DisGCMAE against five categories of methods: Traditional Machine Learning Methods (SVM, MLP, LSTM), GNN-based Models (GCN, GFormer, Hyper-GCN (Feng et al., 2019)), EEG-specific Models (EEGNet (Lawhern et al., 2018), DGCNN, EEG-Conformer (Song et al., 2022), RGNN), Graph Contrastive Pre-training Models (GCC, GraphCL, GRACE, AutoGCL (Yin et al., 2022)), and Graph Generative Pre-training Models (GraphMAE, GPT-GNN, GraphMAE2, S2GAE (Tan et al., 2023)). As demonstrated in Table 1, our model outperforms all other state-of-the-art methods. Notably, pre-training-based models, including those based on GCL-PT (GCC, GraphCL, GRACE, AutoGCL (Yin et al., 2022)) and GMAE-PT (GraphMAE, GPT-GNN, GraphMAE2, S2GAE (Tan et al., 2023)), utilize large Graph Transformers as their backbone in this study. In contrast, our method can be suitable to both spatial-based Graph Transofrmer and spectral-based vanilla GCNs (DGCNN) as the backbone. We evaluated both tiny and large model sizes. As illustrated in Fig. 2 (a), our tiny model, with only 1.3M parameters, performs comparably to pre-training-based methods with larger models (5.7M parameters). Moreover, our large-tiny model, despite having a similar parameter size to others, significantly outperforms them by about 5% in both AUROC and accuracy. This demonstrates that our approach achieves a superior balance between performance and efficiency, delivering high performance with a more compact parameter set. As illustrated in Fig. 2 (b), we investigated the relationship between model parameters and performance across three factors: model size, model type, and varying input EEG densities. It is evident that when the model type and input EEG density are fixed, the large-size model outperforms the tiny-size model. For a given model, reducing the input density (i.e., using LD data) leads to a decline in performance compared to using HD data. However, after pre-training and distillation, the performance of the initially less effective tiny-size model improves significantly, reaching a level comparable to that of the large-size teacher model using HD data without pre-training. This demonstrates that our proposed GCMAE-PT and GTD loss can enhance model performance while maintaining a lightweight parameter set without compromising efficiency.

### 4.3 Ablation Study Analysis

#### 4.3.1 Electrode density and model size

Table 2 presents ablation experiments examining EEG graphs with varying densities (HD/MD/LD) and model types (teacher/student) with different sizes (tiny/large). The results reveal that as electrode density decreases, performance on EEG recognition tasks deteriorates. The decline is more pronounced when reducing density from MD to LD than from HD to MD. This is because, while the reduction from HD to MD removes redundant electrodes, MD still retains essential information, preserving performance. However, reducing from MD to LD results in the loss of critical electrodes, leading to a significant performance drop. Additionally, ablation experiments comparing different model sizes, including tiny and large versions of the spatial-based graph transformer and spectral-

Table 1: Performance comparison of different methods on two clinical EEG datasets for different classification tasks. Our teacher and student model can adopt both spectral-based GCNs (DGCNN) or spatial-based Graph Transformer as the backbone, whereas other graph pre-training models utilize large Graph Transformers. The experiments encompass both high-density and low-density EEG scenarios. Metrics are reported as AUROC(%)/ACC(%).

| Method | HBN MDD | | HBN ASD | | EMBARC Sex | | EMBARC Severity | |
|---|---|---|---|---|---|---|---|---|
| | HD | LD | HD | LD | HD | LD | HD | LD |
| SVM | 72.5/75.6 | 68.1/70.8 | 56.3/59.6 | 54.8/59.3 | 65.2/68.8 | 68.5/68.2 | 59.4/62.8 | 60.1/60.4 |
| MLP | 73.2/75.7 | 71.6/72.5 | 58.3/61.1 | 56.3/59.4 | 68.0/71.3 | 65.7/67.4 | 61.5/63.7 | 59.4/62.6 |
| LSTM | 76.7/79.2 | 73.7/76.8 | 60.3/64.6 | 58.4/61.8 | 69.0/71.8 | 67.3/69.3 | 62.8/66.0 | 61.2/63.8 |
| GCN | 75.8/77.6 | 72.3/76.4 | 60.5/63.7 | 59.2/61.8 | 69.1/72.8 | 66.7/69.6 | 63.5/66.2 | 60.7/63.3 |
| GFormer | 80.4/83.6 | 76.3/80.4 | 62.7/64.2 | 61.5/62.8 | 71.8/74.4 | 68.1/71.6 | 66.2/69.8 | 64.7/66.8 |
| Hyper-GCN | 77.6/80.8 | 75.4/77.7 | 60.1/64.5 | 59.7/63.1 | 70.5/73.8 | 67.6/72.3 | 64.7/68.3 | 63.0/66.2 |
| EEGNet | 80.6/82.9 | 76.3/80.1 | 62.0/64.6 | 59.3/62.8 | 71.1/74.0 | 66.6/70.3 | 65.4/70.1 | 62.4/65.2 |
| EEG Conformer | 79.3/83.1 | 77.5/79.8 | 61.6/64.3 | 60.3/62.4 | 72.2/74.8 | 67.9/70.7 | 64.7/66.3 | 64.2/64.8 |
| RGNN | 79.4/82.5 | 76.8/79.2 | 60.3/62.6 | 58.4/63.2 | 71.8/73.5 | 68.7/71.5 | 64.7/66.2 | 62.5/65.1 |
| DGCNN | 77.1/81.7 | 74.2/78.7 | 61.3/63.8 | 59.3/62.7 | 70.6/73.2 | 66.6/72.3 | 65.4/67.8 | 62.7/64.2 |
| GCC | 82.2/85.1 | 80.4/82.8 | 64.3/66.1 | 63.2/62.1 | 72.9/75.7 | 69.5/73.7 | 68.5/71.1 | 66.1/68.7 |
| GRACE | 83.7/84.6 | 79.9/81.8 | 63.7/66.8 | 61.6/63.9 | 73.2/74.9 | 70.7/72.8 | 67.3/71.8 | 66.7/67.6 |
| GraphCL | 81.7/83.9 | 78.6/80.6 | 64.6/65.4 | 63.4/64.1 | 72.8/75.4 | 68.5/73.5 | 69.4/72.6 | 67.3/69.2 |
| AutoGCL | 82.5/83.5 | 79.3/80.1 | 63.2/65.7 | 61.3/62.2 | 73.1/75.2 | 70.5/73.3 | 69.2/71.4 | 66.3/68.2 |
| GraphMAE | 82.8/85.3 | 79.5/83.3 | 65.1/64.7 | 62.5/62.9 | 72.6/76.3 | 70.2/73.8 | 69.4/69.8 | 65.8/68.5 |
| GPT-GNN | 83.3/85.2 | 80.7/82.2 | 65.6/66.9 | 64.3/65.0 | 71.2/74.7 | 68.5/72.9 | 68.3/70.4 | 65.4/67.1 |
| GraphMAE2 | 83.5/85.7 | 81.3/83.0 | 65.3/65.9 | 62.5/65.9 | 72.2/75.6 | 69.5/73.2 | 70.5/70.0 | 66.2/68.3 |
| S2GAE | 83.0/82.2 | 81.4/79.8 | 66.7/65.3 | 65.7/63.1 | 71.8/73.6 | 68.6/70.4 | 68.3/71.2 | 64.6/69.2 |
| Ours-Tiny (DGCNN) | 84.8/85.4 | 81.6/82.4 | 66.1/66.4 | 63.4/64.1 | 73.4/76.7 | 71.8/75.6 | 68.6/71.9 | 66.8/69.3 |
| Ours-Large (DGCNN) | 86.6/87.4 | 84.4/85.3 | 67.3/68.8 | 66.7/65.9 | 75.4/77.8 | 74.5/76.3 | 71.5/74.6 | 69.2/72.8 |
| Ours-Tiny (Gformer) | 85.3/86.8 | 82.6/84.3 | 66.6/67.8 | 64.7/65.7 | 75.2/77.1 | 73.3/75.3 | 68.7/73.5 | 67.3/72.1 |
| Ours-Large (Gformer) | 87.4/87.8 | 84.8/86.9 | 68.6/69.4 | 66.8/67.4 | 76.6/77.9 | 75.7/76.8 | 72.3/77.2 | 70.6/74.0 |

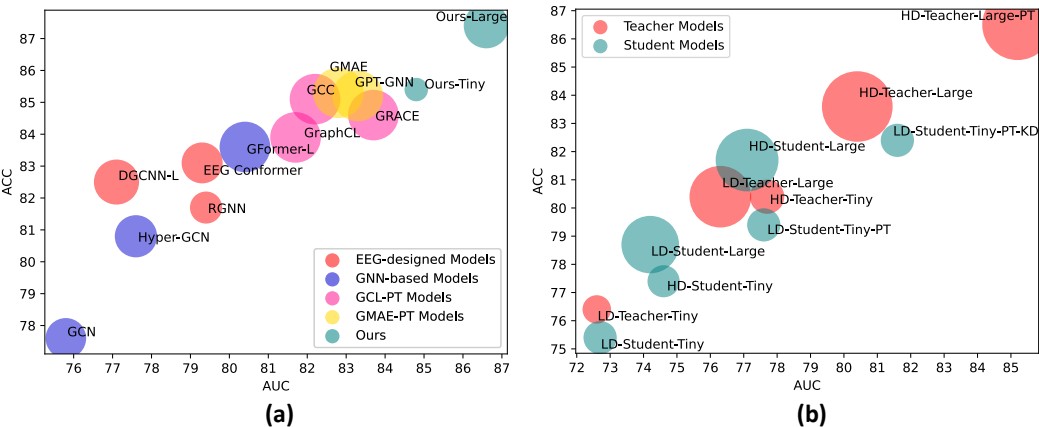

Figure 2: Figure (a) compares model sizes and performance and figure (b) compares model sizes and performance among different types of models. Both (a) and (b) were conducted on HBN for MDD classification task. 'L' denotes large-size models. Models with the same color represent the same type of model, and the size of the circle indicates the number of parameters in the model, with larger circles indicating more parameters. In Fig. (b), the model backbone is vanilla GCNs (DGCNN).

based DGCNN, indicate that the teacher model consistently outperforms the student model of the same size. The tiny teacher model performs similarly to the large student model, and within the same model type, the large model substantially exceeds the tiny model in performance.

### 4.3.2 ANALYSIS OF DIFFERENT PRE-TRAINING METHODS

As detailed in Table 3, we compared our GCMAE-PT with three other pre-training approaches: graph contrastive pre-training (GCL-PT) (You et al., 2020), graph masked autoencoder pre-training (GMAE-PT) (Hou et al., 2022), and a sequential combination of GCL-PT and GMAE-PT (Seq. Comb.). Following pre-training, we evaluated the models on downstream classification tasks. The results indicate that our framework surpasses GCL-PT, GMAE-PT, and their sequential combination. This underscores that sequentially combining contrastive and generative pre-training methods does

Table 2: The ablation explores the impact of varying EEG densities (HD/MD/LD), model types (teacher/student), and sizes (tiny/large) on performance. 'T' denotes teacher models and 'S' denotes student models. The experiments were conducted on the HBN dataset for the MDD classification task, with all models evaluated without pre-training. Metrics are reported as AUROC(%)/ACC(%).

| Density | GFormer-Large (T) | | Gformer-Tiny (S) | |
|---|---|---|---|---|
| | Tiny | Large | Tiny | Large |
| LD | 72.6/76.4 | 76.3/80.4 | 72.7/75.4 | 74.2/78.7 |
| MD | 75.6/78.1 | 78.7/82.5 | 74.3/77.2 | 76.0/80.5 |
| HD | 77.7/80.4 | 80.4/83.6 | 75.6/78.4 | 77.1/81.7 |

Table 3: Ablation studies were conducted on our GCMAE-PT, comparing it with GCL-PT, GMAE-PT, and their sequential combination (Seq. Comb.). 'T' and 'S' represent the teacher and student models, respectively. The experiments were performed on the HBN and EMBARC datasets. Baseline results can be found in Table 2. The teacher model uses HD inputs with a large-size configuration, while the student model uses LD inputs with a tiny-size configuration.

| GPT Methods | HBN MDD | | EMBARC Sex | |
|---|---|---|---|---|
| | HD-T-Large | LD-S-Tiny | HD-T-Large | LD-S-Tiny |
| Baseline | 80.4/83.6 | 72.7/75.4 | 71.8/74.4 | 64.2/69.3 |
| GCL-PT | 82.2/85.1 | 74.3/75.9 | 72.9/75.7 | 66.4/71.5 |
| GMAE-PT | 82.8/85.3 | 75.1/76.6 | 72.6/76.3 | 67.1/71.7 |
| Seq. Comb. | 83.3/85.9 | 75.6/78.1 | 73.3/77.1 | 67.6/72.5 |
| GCMAE-PT | **85.2/86.5** | **77.6/79.4** | **74.7/78.4** | **68.8/73.3** |

not achieve optimal performance. Our approach, which seamlessly integrates these techniques into a cohesive framework with explicit and implicit mutual supervision, delivers superior results.

### 4.3.3 ANALYSIS OF EEG PATTERNS FOR DIFFERENT MASKING AND RECONSTRUCTION

To illustrate the effectiveness of our proposed pre-training method, we visualized EEG data patterns across various densities, masking ratios, and reconstruction methods, as depicted in Fig. 3. Figure 3(a) shows clear and well-connected activated regions with no masking. As we increased the masking ratio in Figures 3(b), 3(c), and 3(d), the activated regions diminish and connectivity deteriorates, reflecting increased information loss. Figure 3(e) demonstrates the effectiveness of our reconstruction method with 50% masking, revealing a pattern that closely resembles the unmasked data in Fig. 3(a), with improved activation and high reconstruction accuracy.

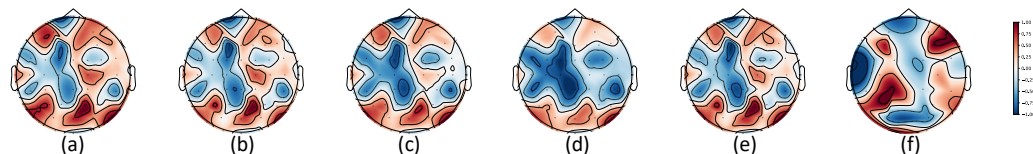

Figure 3: Ablation studies of EEG patterns on the EMBARC datasets for MDD severity classification task. (a) is the pattern of original HD EEG w/o masking. (b), (c) and (d) are patterns of HD EEG w/ 25%, 50% and 75% masking ratios and reconstructed by vanilla GMAE-PT, respectively. (e) is the pattern of HD EEG w/ 50% masking ratio and reconstrcted by our proposed GCMAE-PT. (f) is the pattern of original MD EEG w/o masking.

## 5 ANALYSIS OF (PRE-)TRAINING AND DISTILLATION

As shown in Figure 4, we visualized the optimization process of the loss curves, including contrastive loss, reconstruction loss, and GTD loss, during both the pre-training and fine-tuning stages. Figure 4(a) shows that during pre-training, we jointly optimized the contrastive loss and reconstruction loss for both the teacher and student models. All four losses converge effectively during

Table 4: Ablation studies on logits distill loss and our GTD loss. T and S denote teacher and student. The experiments are conducted on the HBN dataset for MDD classification.

| GKD Methods | w/o Pre-training | | w/ Pre-training | |
|---|---|---|---|---|
| | HD-T-Large | LD-S-Tiny | HD-T-Large | LD-S-Tiny |
| Baseline | 80.4/83.6 | 72.7/75.4 | 85.2/86.5 | 77.6/79.4 |
| + Logits (Hinton et al., 2015) | - | 73.6/77.3 | - | 79.6/80.7 |
| + Proposed | - | 73.8/78.5 | - | 80.2/81.5 |
| + Union | - | **75.0/79.2** | - | **81.6/82.4** |
| + LSP (Yang et al., 2020) | - | 73.1/76.7 | - | 78.8/80.7 |
| + G-CRD (Joshi et al., 2022) | - | 73.4/77.8 | - | 79.3/80.3 |

optimization. Notably, the contrastive loss for both the teacher and student models exhibit similar optimization trends, as do the reconstruction losses. Figure 4(b) illustrates the impact of the proposed GCMAE-PT and GTD loss on downstream classification tasks. We present the optimization curves for the general Cross-Entropy (CE) loss, as well as the optimization curves after applying GCMAE-PT, GTD loss, and both combined. It is clear that the CE loss is better optimized with the application of GCMAE-PT and GTD loss. This confirms that both pre-training with GCMAE-PT and the use of GTD loss enhance the performance of downstream classification tasks.

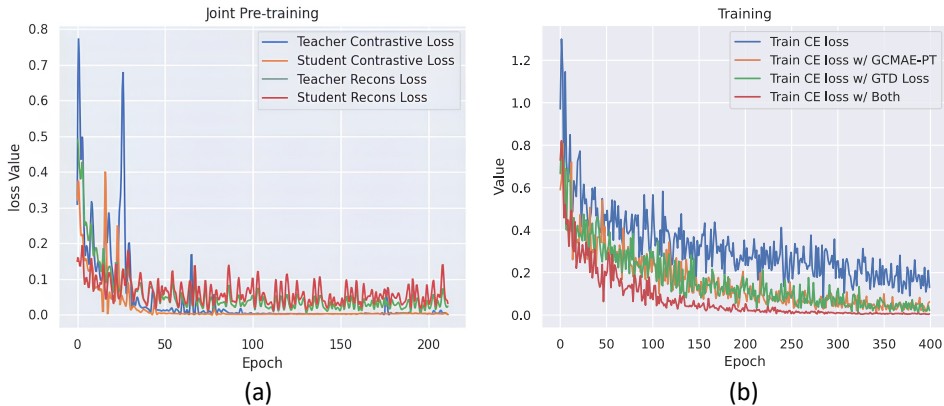

(a)                                    (b)

Figure 4: Illustrations of loss curves in both the pre-training stage (a) and fine-tuning stage (b). We applied early stopping to prevent overfitting. This also indicates that our GTD loss effectively accelerates convergence and helps avoid overfitting.

## 6 CONCLUSION

In this paper, we present an innovative framework for EEG pre-training and distillation, which effectively integrates contrastive-based and generative-based graph pre-training paradigms. Furthermore, our framework incorporates a specifically designed EEG graph topology distillation loss function, tailored for the distillation process from high-density to low-density EEG data. Our method demonstrates substantial and efficient improvements over contemporary approaches, significantly enhancing the accuracy of EEG-based disease diagnosis while facilitating seamless deployment across diverse medical devices. Moreover, our method is readily extendable to a range of EEG application scenarios, including emotion recognition, brain-computer interfacing, and epilepsy detection.

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

APPENDIX

## A  IMPLEMENTATION DETAILS

All our training was conducted on an NVIDIA GeForce RTX 4090 GPU. During pre-training, we used a batch size of 128, trained for 200 epochs. For downstream fine-tuning, we used a batch size of 32, trained for 400 epochs. Both pre-training and fine-tuning were optimized using the Adam optimizer.

## B  PRELIMINARIES OF DYNAMIC GNNS

In traditional GNNs, the adjacency matrix $\mathcal{A}$ is static. However, in this paper, we adopt dynamic GNNs, where the adjacency matrix can be dynamically adjusted during training to suit the specific task better. This approach allows the model to adapt the graph structure based on the input data and learning objectives. In such models, the edge weights $\alpha_{ij}$ between nodes $(i, j)$ are learned during training. The edge weights can be computed as:

$$\alpha_{ij} = \sigma\left(f\left(\mathcal{X}_i, \mathcal{X}_j\right)\right) \tag{15}$$

where $f(\cdot)$ is a function for calculating edge weights, and $\sigma$ is an activation function (e.g., Sigmoid). The dynamic adjacency matrix $\mathcal{A}$ is then updated based on these weights, typically using a thresholding mechanism:

$$\mathcal{A}_{ij} = \begin{cases} 1, & \text{if } \alpha_{ij} > \theta \\ 0, & \text{otherwise} \end{cases} \tag{16}$$

where $\theta$ is a threshold. During message passing, the dynamic adjacency matrix influences how messages are aggregated from neighboring nodes:

$$\mathbf{m}_i = \sum_{j \in \mathcal{N}(i)} \alpha_{ij} \mathbf{W} x_j \tag{17}$$

Here, $\alpha_{ij}$ represents the dynamically computed edge weight used to weight the messages from neighbors. Node features are updated as follows:

$$\mathcal{X}_i^{(l+1)} = \sigma\left(\mathbf{W}^{(l)} \mathcal{X}_i^{(l)} + \mathbf{b}^{(l)} + \mathbf{m}_i\right) \tag{18}$$

By dynamically adjusting the adjacency matrix, dynamic GNNs can capture more complex and evolving relationships within the graph, thereby enhancing flexibility and overall performance.

## C  DETAILS OF MOTIVATION AND PROBLEM

### C.1  GTL FOR UNLABELED/LABELED EEG

Many existing methods primarily focus on training models with limited labeled EEG data, overlooking the potential of abundant unlabeled data. These methods emphasize novel GNN architectures but fail to fully leverage the available data. Additionally, they do not exploit high-density (HD) EEG data to improve models for low-density (LD) scenarios. This underscores the need for strategies that integrate both labeled and unlabeled data, and use HD data to enhance performance in LD contexts.

Moreover, most pre-training methods are directly applied to EEG time series, with very few addressing the issue from the perspective of large-scale graph pre-training. In contrast, our approach proposes pre-training EEG graph models using a graph-based pre-training perspective. This not only aims to transfer knowledge from unlabeled EEG data to tasks on labeled EEG data but also benefits HD-to-LD distillation. This is based on the following observation:

***Observation:*** An LD EEG graph can be viewed as an HD EEG graph with specific nodes removed. In graph contrastive self-supervised pre-training, contrastive views are obtained by graph augmentation, such as removing nodes and edges. Another graph pre-training method, graph masked autoencoders pre-training, operates by masking node features and then reconstructing them. The relationships between these methods are formulated as follows:

$$\underbrace{Density\ Decrease}_{Electrodes\ Loss} \Longleftrightarrow \underbrace{Node\ Dropping}_{GCL\ Augmentation} \Longleftrightarrow \underbrace{Node\ Masking}_{GMAE\ Masking} \tag{19}$$

Based on this observation, we propose a novel unified graph self-supervised pre-training paradigm called GCMAE-PT. This approach intricately combines Graph Contrastive Pre-training (Qiu et al., 2020; You et al., 2020) with Graph Masked Autoencoders Pre-training (Hou et al., 2022), allowing us to model and capture the relationships among the three entities described in Eq. 20.

### C.2 GKD FOR HIGH/LOW-DENSITY EEG

As previously mentioned, an LD EEG graph can be viewed as an HD EEG graph with specific nodes removed. Consequently, HD EEG contains many features that LD EEG lacks. We naturally formulate this as a graph knowledge distillation (GKD) task, focusing on how to transfer information from HD EEG data to LD EEG applications, which is a data-level distillation process. Additionally, if a more complex teacher model with a larger number of parameters is used to extract features from HD EEG data, and a simpler student model with fewer parameters is used for LD EEG data, this involves model-level distillation. The aim is to deploy the lightweight student model while ensuring that its performance approaches, or even surpasses, that of the more cumbersome teacher model.

Therefore, the GKD process can be represented by the following formula:

$$\underbrace{Teacher\ Model}_{HD\ EEG\ Data} \xrightarrow[Distill\ (Data\text{-}level)]{Compress\ (Model\text{-}level)} \underbrace{Student\ Model}_{LD\ EEG\ Data} \tag{20}$$

## D DATA COLLECTION AND PRE-PROCCESSING

### D.1 EEG DATA QUANTITY STATISTICS

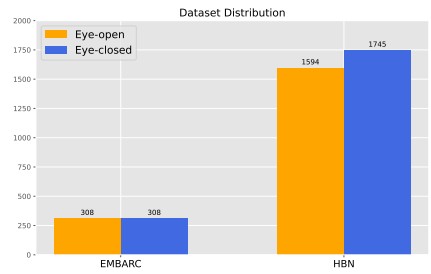

Figure 5: Illustration of statistical data distribution in the EMBARC and HBN datasets.

As illustrated in Fig. 5, the EMBARC dataset consists of EEG signals collected from 308 subjects in both eye-open and eye-closed states. The EEG time series were sampled at 250Hz, with each trial lasting approximately 200 seconds. Similarly, in the HBN dataset, EEG signals were collected in both eye-open and eye-closed states, with 1,594 subjects for the eye-open condition and 1,764 subjects for the eye-closed condition. The duration of the recordings is also around 200 seconds, with the same sampling frequency of 250Hz. Both EMBARC and HBN datasets use the 10-20 EEG standard system, with EMBARC employing a 64-electrode cap and HBN using a 128-electrode cap.

### D.2 EXPLANATION OF UNLABELED DATA

Collecting EEG recordings, each patient diagnosed with a particular mental disorder can be classified as a labeled subject. Patients with EEG diagnosed as other disorders or healthy controls, are categorized as unlabeled data. In clinical, the amount of labeled data diagnosed as certain disorders was limited. Therefore, models trained exclusively on such sparse labeled data are prone to underfitting, undermining their predictive performance. However, by broadening the scope to include aggregated data from a range of disorders to form a comprehensive unlabeled or mixed-labeled dataset, pre-training models on this enriched dataset can mitigate the constraints imposed by data scarcity. This approach enhances the model's generalizability and improves performance, even in the face of limited labeled examples.

### D.3 CONSTRUCTION AND AUGMENTATION OF PRE-TRAINING GRAPH DATASETS

To construct the pre-training dataset, we combined the data from both the eye-open and eye-closed states from these two datasets. For EEG data augmentation, we applied a sliding window sampling

method for each subject in the EMBARC and HBN datasets. EEG time series segments were extracted every 50 seconds, with a 20-second overlap between consecutive segments. The formula for calculating the number of segments for each subject is as follows:

$$Segments\ per\ Subject = \left\lfloor \frac{Total\ length - Window\ length}{Window\ length - Overlap\ length} \right\rfloor + 1 \qquad (21)$$

Additionally, we combined the entire time series for each subject with the extracted segments. For each time series segment, we computed the Power Spectral Density features and then constructed the EEG graph samples. The formula for calculating the total number of samples used in the construction of the pre-training datasets is as follows:

$$Total\ Samples = (Segments\ per\ Subject + 1) \times Subjects \qquad (22)$$

Note that the term *Subjects* here refers to the combination of EEG segments from both EMBARC and HBN datasets, including both eye-open and eye-closed EEG samples. Ultimately, we obtain approximately 4,000 samples ($308 + 308 + 1,594 + 1,764 \approx 4,000$), resulting in about 24,000 EEG graph samples for the graph self-supervised pre-training corpus.

### D.4 (PRE-)TRAINING AND EVALUATION SETTINGS

For pre-training on the EMBARC dataset, we addressed the issue of dataset size disparity between EMBARC and the HBN dataset, which both originate from the same EEG system. Specifically, we downsampled the 128 channels of the HBN data to 64, 32, and 16 channels, maintaining the same arrangement. These downsampled data were then combined with the corresponding density datasets from EMBARC to create a unified pre-training dataset. Note that, as the EMBARC dataset does not include 128 channels, the 128-channel HD pre-training dataset does not incorporate data from EMBARC (HBN only).

For downstream task fine-tuning, due to the limited amount of labeled data, we employed 10-fold cross-validation with 10 runs for all model training. The Adam optimizer (Kingma, 2014) was used to optimize the training process. Pre-training was performed over 200 epochs, while downstream fine-tuning was carried out for 400 epochs.

### D.5 CONSTRUCTION OF DOWNSTREAM DATASETS

Table 5 provides the quantity of labeled data for four downstream classification tasks across the EMBARC and HBN datasets.

In the EMBARC dataset, the number of subjects is consistent across eye-open and eye-closed conditions. For the MDD sex classification task, there are 296 subjects with varying levels of depression (all diagnosed with depression) and 12 normal subjects. Among the depressed individuals, there are 194 males and 102 females. For the depression severity classification task, 166 subjects are diagnosed with severe depression ($HAMD_{17}$ score $> 17$) Boessen et al. (2013), and 130 subjects are diagnosed with mild depression ($HAMD_{17}$ score $\leq 17$).

The HBN dataset, which includes a range of diseases, has significantly fewer labeled samples compared to the total data volume due to the high number of samples without explicit MDD and ASD diagnostic labels. Additionally, the number of labeled subjects differs between eye-open and eye-closed conditions. In the eye-open data, there are 178 healthy controls, 109 MDD patients, and 234 ASD patients. In the eye-closed data, there are 187 healthy controls, 120 MDD patients, and 245 ASD patients.

To ensure a large-scale pre-training dataset, we utilized slicing operations to expand the dataset size. However, for constructing labeled datasets for downstream tasks, slicing was not employed. Instead, we calculated the PSD features for the entire 200-second EEG time series.

### D.6 COMPARISON BETWEEN THE PRE-TRAINING DATASET AND DOWNSTREAM DATASETS

For the pre-training dataset, which includes both labeled and unlabeled data, we applied slicing operations to significantly increase the dataset size. In contrast, for the downstream dataset, particularly for the HBN data, the labeled data constitutes only a small fraction of the total dataset, and no

slicing operations were performed. In this context, it is crucial to leverage the pre-training dataset effectively to enhance model performance on the limited labeled data available.

Table 5: Labeled data distribution of the EMBARC and HBN datasets. 'HC' means healthy control.

| Datasets | EMBARC | | HBN | |
| | Sex | Severity | MDD | ASD |
|---|---|---|---|---|
| Eye-open | Female: 194 Male: 102 | Severe: 166 Mild: 130 | Patient: 109 HC: 178 | Patient: 234 HC: 178 |
| Eye-closed | Female: 194 Male: 102 | Severe: 166 Mild: 130 | Patient: 120 HC: 187 | Patient: 245 HC: 187 |

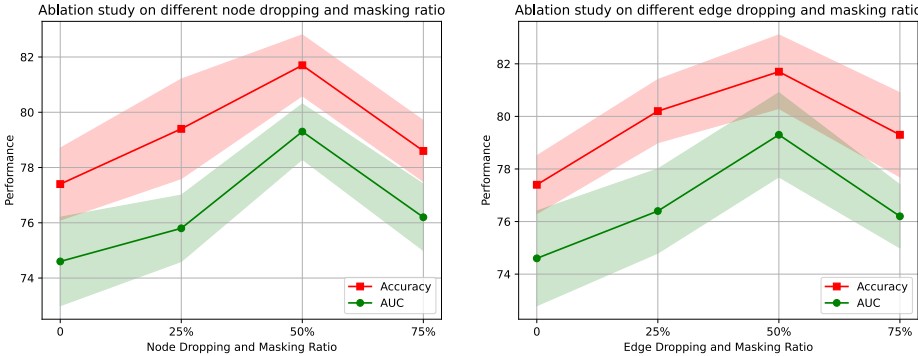

Figure 6: Ablation studies on different node and edge dropping (for GCL-PT) and masking (for GMAE-PT) ratios. A 50% masking ratio for both nodes and edges achieves the best performance.

# E    DIFFERENT CONFIGURATIONS OF TEACHER AND STUDENT MODELS

Table 6: Comparison of Model Configurations. Note that DGCNN is a **spectral-based** vanilla GCNs model (DGCNN), while GFormer means the **spatial-based** Graph Transformer model. We considered both types of graph models to demonstrate the versatility of our pipeline.

| Model | Encoder | Sizes (S) | Layers (L) | Dimensions (D) | Heads (H) | Position Embedding (P) | Params (PM) |
|---|---|---|---|---|---|---|---|
| Teacher | DGCNN | Large | 8 | 128 | - | ✗ | 5.7M |
| | GFormer | Large | 8 | 128 | 8 | ✓ | 6.9M |
| Student | DGCNN | Tiny | 4 | 64 | - | ✗ | 1.3M |
| | GFormer | Tiny | 4 | 64 | 4 | ✓ | 1.4M |

# F    ABLATION STUDY ON DIFFERENT EEG BANDS

As shown in Table 7, we performed ablation studies on various EEG frequency bands across four tasks on two datasets and observed that the alpha band consistently yielded the best performance across all tasks. Consequently, we selected the alpha band as our primary configuration.

# G    ALGORITHM PIPELINE OF GTD LOSS

To clarify the GTD loss calculation, we present the pipeline as shown in Algorithm 1.

Table 7: Ablation experiments on performance of different EEG bands. The model employed is a **tiny-sized student** model obtained through pre-training and distillation with **LD EEG input**.

| Datasets | Downstream Tasks | Alpha | Beta | Gamma | Theta |
|---|---|---|---|---|---|
| HBN | MDD Diagnosis | 84.8/85.4 | 82.6/84.1 | 81.3/82.5 | 79.6/80.6 |
| | ASD Diagnosis | 66.1/66.4 | 61.4/64.1 | 63.3/63.5 | 60.7/62.6 |
| EMBARC | MDD Sex | 73.4/76.7 | 71.6/72.4 | 70.3/70.9 | 68.5/72.9 |
| | MDD Severity | 68.6/71.9 | 66.3/67.2 | 64.5/66.8 | 63.7/66.2 |

---

**Algorithm 1: GTD Loss Calculation**

---

**Input**: $\mathcal{X}^l, \mathcal{V}^l, \mathcal{A}^l, \mathcal{X}^h, \mathcal{V}^h, \mathcal{A}^h, \mathcal{V}^d$
**Parameter**: $\mathcal{F}(.), \theta, \epsilon$
**Output**: $\mathcal{L}_{Dis}^{GTD}$

1: Normalize $\mathcal{A}^l, \mathcal{A}^h$
2: Apply threshold: $\mathcal{A}^l \leftarrow (\mathcal{A}^l > \theta), \mathcal{A}^h \leftarrow (\mathcal{A}^h > \theta)$
3: Compute kernel matrices: $\mathcal{Z}^l = \mathcal{F}(\mathcal{X}^l), \mathcal{Z}^h = \mathcal{F}(\mathcal{X}^h)$
4: **Assert:** $|\mathcal{V}^l| \leq |\mathcal{V}^h|$
5: **if** $|\mathcal{V}^l| \neq |\mathcal{V}^h|$ **then**
6:     Extract sub-matrices $\mathcal{A}_{sub}^h = \mathcal{A}^h[\mathcal{V}^l, \mathcal{V}^l]$
7:     Direct connections: $\mathcal{A}_{1-hop}^h = (\mathcal{A}_{sub}^h > 0)$
8:     Indirect connections: $\mathcal{A}_{2-hop}^h = \mathcal{A}^h[\mathcal{V}^d, : \mathcal{V}^l]$
9:     $\mathcal{L}_{Pos} = KL(\mathcal{Z}^l||\mathcal{Z}^h) \,|\, (\mathcal{A}_{1-hop}^h \vee \mathcal{A}_{2-hop}^h)$
10:    $\mathcal{L}_{Neg} = KL(\mathcal{Z}^l||\mathcal{Z}^h) \,|\, (\mathcal{A}^l \wedge \neg(\mathcal{A}_{1-hop}^h \vee \mathcal{A}_{2-hop}^h))$
11: **else**
12:    $\mathcal{L}_{Pos} = KL(\mathcal{Z}^l||\mathcal{Z}^h) \,|\, (\mathcal{A}^h > 0)$
13:    $\mathcal{L}_{Neg} = KL(\mathcal{Z}^l||\mathcal{Z}^h) \,|\, (\mathcal{A}^l > 0 \wedge \mathcal{A}^h = 0)$
14: **end if**
15: $\mathcal{L}_{PosAvg} = \frac{\mathcal{L}_{Pos}}{C_{pos}}$
16: $\mathcal{L}_{NegAvg} = \frac{\mathcal{L}_{Neg}}{C_{neg}}$
17: **return** $\mathcal{L}_{Dis}^{GTD} = \frac{\mathcal{L}_{PosAvg}}{\mathcal{L}_{NegAvg}+\epsilon}$

---

## H   SIMILARITY KERNEL SELECTION FOR GTD LOSS

We follow (Joshi et al., 2022) and try different similarity kernels to measure the distance between nodes. All four kernels are shown in the following formula:

$$
\begin{aligned}
\textit{Linear Kernel:} &\quad \mathcal{Z}_{ij} = \mathcal{F}(\mathcal{X}_i, \mathcal{X}_j) = \mathcal{X}_i \cdot \mathcal{X}_j \\
\textit{Euclidean Kernel:} &\quad \mathcal{Z}_{ij} = \mathcal{F}(\mathcal{X}_i, \mathcal{X}_j) = \|\mathcal{X}_i - \mathcal{X}_j\|_2 \\
\textit{Polynomial Kernel:} &\quad \mathcal{Z}_{ij} = \mathcal{F}(\mathcal{X}_i, \mathcal{X}_j) = (\mathcal{X}_i \cdot \mathcal{X}_j + c)^d \\
\textit{RBF Kernel:} &\quad \mathcal{Z}_{ij} = \mathcal{F}(\mathcal{X}_i, \mathcal{X}_j) = \exp(-\gamma\|\mathcal{X}_i - \mathcal{X}_j\|_2^2)
\end{aligned}
\tag{23}
$$

As shown in Fig. 7, we conducted ablation experiments on the GTD loss. Figure 7(a) illustrates the results of distillation in three scenarios: high-to-low (H2L), high-to-medium (H2M), and high-to-high (H2H). Note that H2H is a special case. Although the GTD loss is designed primarily for high-to-low density distillation, it can also be applied to high-to-high density distillation as an exception.

The optimization curves for H2L and H2M show good convergence. However, in the special case of H2H, while the optimization curve also converges, the gradient descent is less pronounced. This suggests that although GTD loss can still be applied in the H2H scenario, it is less effective. This is because GTD loss mainly focuses on nodes that are removed, and since no nodes are removed in H2H, the distillation's primary goal is to correct the student model's misinterpretation of connectivity. Consequently, there is less knowledge to distill compared to H2L and H2M scenarios, resulting in a less noticeable decrease in the optimization curve. In contrast, the optimization curve for H2L shows the most significant decrease, followed by H2M.

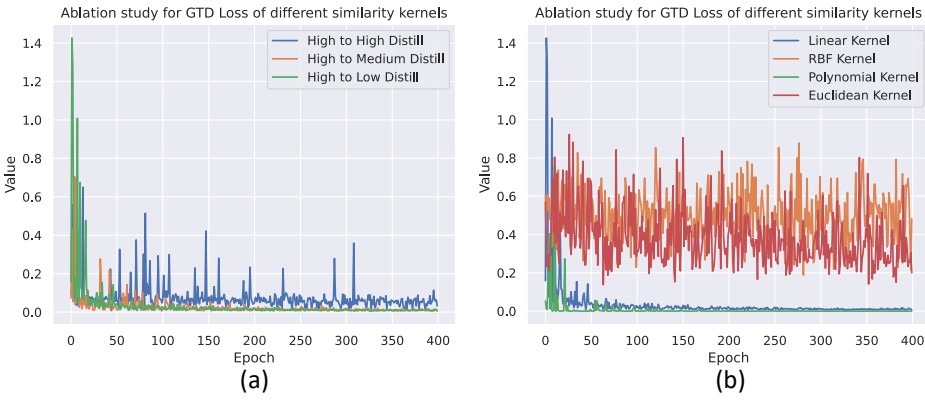

Figure 7: The ablation studies of distillation across different density settings (a) and kernels (b).

Figure 7(b) presents the results of ablation experiments with different similarity kernels. The experimental results indicate that only the GTD loss using polynomial and linear kernels achieved good convergence during optimization. Among these, the linear kernel provided the best distillation effect, which is why we selected it as the primary kernel for our GTD loss.

## I    CLINICAL INTERPRETATION OF EEG PATTERNS

By grounding our visual findings in these established studies, we provide a clearer link between the reconstructed EEG patterns and their clinical implications, emphasizing the robustness and diagnostic utility of our approach.

To address the clinical relevance of our EEG pattern reconstructions, we link the visual patterns presented in Figure 3 to established clinical findings in MDD research. The unmasked EEG pattern in Figure 3(a) reveals clear activations in the frontal and central regions, which are crucial areas involved in cognitive processing and emotional regulation. These regions are frequently highlighted in MDD studies due to their role in mood and executive function. Specifically, the prefrontal cortex, anterior cingulate cortex, and related regions are implicated in emotional processing and regulation, with MDD patients often showing disrupted activity in these areas (Davidson et al., 2002; Price & Drevets, 2012). Reduced activation in these areas can reflect difficulties in cognitive control and emotional regulation, key features of depressive symptoms.

As the masking ratio increases (Figures 3b-d), the patterns show a noticeable decline in activation and connectivity, particularly in the frontal and central regions. This aligns with findings in MDD literature, where disrupted functional connectivity, especially in the frontocingulate networks, is a well-documented feature of the disorder. For example, alterations in prefrontal connectivity are often associated with the severity of depressive symptoms and the inability to regulate negative emotions (Pizzagalli, 2011). The degradation observed in Figures 3b-d is consistent with the hypothesis that higher masking ratios simulate information loss, highlighting the importance of intact frontal connectivity for accurate MDD classification.

Critically, Figure 3(e), which displays the reconstructed pattern using our proposed GCMAE-PT with 50% masking, closely resembles the unmasked pattern seen in Figure 3(a). The reconstructed data retain key activations in the frontal and parietal regions, indicating that our method effectively preserves clinically relevant EEG features even under challenging conditions. This preservation is crucial because altered activity in these regions, particularly in the alpha and theta bands, is often linked to cognitive and emotional dysregulation in MDD patients (Thibodeau et al., 2006; Knyazev, 2007). For instance, lower alpha activity in the frontal regions has been associated with greater emotional dysregulation, while changes in theta activity are linked to altered cognitive processes, both of which are core characteristics of MDD.

The preserved patterns in Figure 3(e) suggest that GCMAE-PT can maintain these clinically significant EEG characteristics, which are essential for accurate classification of MDD severity. This find-

ing not only demonstrates the robustness of our reconstruction method but also aligns with known clinical markers of MDD, supporting the practical relevance of our approach. Furthermore, the ability to accurately reconstruct these key patterns contributes directly to classification tasks, as regions showing consistent and clinically significant alterations are critical for distinguishing between different severity levels of MDD. By maintaining the integrity of essential EEG features under masking, our method ensures that the reconstructed data remain informative and diagnostically valuable, potentially leading to better predictive performance in clinical settings.

## J EXPERIMENTS ON VERY-LOW DENSITY

To further test the generalization ability of our model in extreme scenarios, we evaluated it under a very low-density (VLD) condition. Specifically, we tested the extreme case with EEG data using only 8 electrodes. As shown in Table 8, our proposed pre-training framework and the corresponding GTD loss are able to tackle the extreme case with very few electrodes.

Table 8: Experiments on the very low-density (VLD) situation. HD -> LD/VLD means high-density to (very)low-density distillation.

| PT Methods | PT Loss | FT Loss | HD -> LD | | HD -> VLD | |
|---|---|---|---|---|---|---|
| | | | Sex | Severity | Sex | Severity |
| GCL-PT | w/o GTD | w/o GTD | 1.8%↑ | 1.7%↑ | 1.5%↑ | 2.0%↑ |
| GMAE-PT | w/o GTD | w/o GTD | 1.6%↑ | 1.5%↑ | 2.1%↑ | 2.4%↑ |
| GCMAE-PT (Ours) | w/o GTD | w/o GTD | **2.9%↑** | **3.8%↑** | **3.5%↑** | **4.4%↑** |
| GMAE-PT | w/ GTD | w/ GTD | 2.2%↑ | 3.0%↑ | 1.9%↑ | 2.2%↑ |
| GCL-PT | w/ GTD | w/ GTD | 2.1%↑ | 3.2%↑ | 1.5%↑ | 2.0%↑ |
| GCMAE-PT (Ours) | w/ GTD | w/ GTD | **4.8%↑** | **6.4%↑** | **4.3%↑** | **5.2%↑** |

## K EXPERIMENTS OF DIFFERENT FINE-TUNING PARADIGMS

Table 9: Experiments on the effectiveness and efficiency of different fine-tuning (FT) methods. The experiments are conducted on HBN dataset for MDD classification. The unit of fine-tuning speed is seconds (s), and the unit of memory cost is gigabytes (G). The input data consists of 128-channel HD EEG graphs, and the model uses a large-size graph transformer.

| Fine-tuning Methods | Effectiveness | | Efficiency | |
|---|---|---|---|---|
| | AUROC | ACC | FT Speed | Memory Cost |
| Vanilla FT | 80.4% | 83.6% | 183s | 1.0G |
| Parameter-Efficient FT | 78.7% | 83.4% | 86s | 0.3G |

As shown in Table 9, we evaluated two distinct fine-tuning paradigms. The first, termed Vanilla FT, involves fine-tuning all parameters of the pre-trained encoder. The second, referred to as parameter-efficient FT, entails freezing the lower layers of the pre-trained encoder and fine-tuning only the parameters of the upper layers, such as the fully connected layers. It is evident that parameter-efficient FT, which requires fewer parameters to be optimized, results in a fine-tuning speed three times faster and memory usage one-third that of Vanilla FT. However, this approach incurs a slight performance trade-off compared to Vanilla FT.

## L EXPERIMENTS ON HELD-OUT VALIDATION

In the pretraining dataset of the previous experiment, we integrated heterogeneous EEG data from different diseases to pretrain our model. To further validate the reliability of our model, we conducted a held-out validation experiment.

Table 10: Ablation studies on the proposed pre-training framework and the GTD loss. The held-out validation means we pre-train the model only on the HBN dataset and fine-tune the model to the EMBARC dataset. HD -> LD means high-density to low-density distillation. 'All' means HBN + EMBARC. ↑ means performance improvement in terms of accuracy. Note that the GTD loss is applied in both the pre-training and fine-tuning stages. The downstream task is conducted on the EMBARC dataset for MDD severity classification task. The backbone model is the spatial-based Graph Transformer.

| PT Methods | PT Loss | FT Loss | Datasets | | Distill Performance | |
|---|---|---|---|---|---|---|
| | | | Pre-Train | Fine-Tune | HD -> MD | HD -> LD |
| GCL-PT | w/o GTD | w/o GTD | HBN | EMBARC | 1.5%↑ | 1.7%↑ |
| GMAE-PT | w/o GTD | w/o GTD | HBN | EMBARC | 1.7%↑ | 1.5%↑ |
| Seq. Comb. | w/o GTD | w/o GTD | HBN | EMBARC | 2.0%↑ | 2.1%↑ |
| GCMAE-PT (Held-Out) | w/o GTD | w/o GTD | HBN | EMBARC | **3.1%**↑ | **3.0%**↑ |
| GCMAE-PT (Ours) | w/o GTD | w/o GTD | All | EMBARC | **3.7%**↑ | **3.8%**↑ |
| GCL-PT | w/ GTD | w/o GTD | HBN | EMBARC | 1.9%↑ | 2.0%↑ |
| GMAE-PT | w/ GTD | w/o GTD | HBN | EMBARC | 1.7%↑ | 1.8%↑ |
| Seq. Comb. | w/ GTD | w/o GTD | HBN | EMBARC | 2.3%↑ | 2.6%↑ |
| GCMAE-PT (Held-Out) | w/ GTD | w/o GTD | HBN | EMBARC | **3.7%**↑ | **4.1%**↑ |
| GCMAE-PT (Ours) | w/ GTD | w/o GTD | All | EMBARC | **3.9%**↑ | **4.3%**↑ |
| GCL-PT | w/o GTD | w/ GTD | HBN | EMBARC | 2.3%↑ | 2.2%↑ |
| GMAE-PT | w/o GTD | w/ GTD | HBN | EMBARC | 2.2%↑ | 2.5%↑ |
| Seq. Comb. | w/o GTD | w/ GTD | HBN | EMBARC | 2.7%↑ | 3.1%↑ |
| GCMAE-PT (Held-Out) | w/o GTD | w/ GTD | HBN | EMBARC | **4.4%**↑ | **5.0%**↑ |
| GCMAE-PT (Ours) | w/o GTD | w/ GTD | All | EMBARC | **4.7%**↑ | **5.6%**↑ |
| GCL-PT | w/ GTD | w/ GTD | HBN | EMBARC | 3.3%↑ | 3.0%↑ |
| GMAE-PT | w/ GTD | w/ GTD | HBN | EMBARC | 3.1%↑ | 3.2%↑ |
| Seq. Comb. | w/ GTD | w/ GTD | HBN | EMBARC | 3.2%↑ | 3.9%↑ |
| GCMAE-PT (Held-Out) | w/ GTD | w/ GTD | HBN | EMBARC | **5.0%**↑ | **5.7%**↑ |
| GCMAE-PT (Ours) | w/ GTD | w/ GTD | All | EMBARC | **5.6%**↑ | **6.4%**↑ |

# M ANALYSIS OF SHARED KEY POOL QUEUE

The reason for implementing the proposed teacher-student shared key pool queue is that we have two types of original input data: HD and LD EEG graphs. Through the key pool queue, we allow high-density and low-density EEG key samples to share the same gradient update process within a batch. This approach also enables both the teacher and student models to simultaneously capture shared patterns between these two types of data.

# N CHALLENGES, LIMITATIONS, AND FUTURE WORKS

EEG graph self-supervised pre-training offers a promising avenue for leveraging extensive EEG data, paving the way for large-scale graph-based EEG models. Our proposed GCMAE-PT method is well-suited as a pre-training approach for large-scale EEG foundation model. However, a key challenge is unifying data with varying electrode configurations across different EEG systems to address data heterogeneity.

In our study, while constructing a unified EEG pre-training dataset from multiple sources, we faced the constraint of all datasets being from the same EEG system (10-20 system). To standardize the data, we reduced the number of electrodes in datasets with more electrodes to match those with fewer electrodes, creating a unified pre-training dataset. This approach, however, leads to a loss of information from removed electrodes and restricts the use of datasets with fewer electrodes for pre-training on datasets with more electrodes.

Addressing the challenge of integrating EEG data with differing electrode counts from various systems, while preserving electrode precision, is crucial for developing a comprehensive pre-training dataset. Successfully overcoming this issue could enable large-scale graph pre-training and establish a robust EEG graph foundation model, representing a significant advancement in the field.

# O  PSEUDO-CODE OF GCMAE-PT

To facilitate understanding of our GCMAE-PT, we present PyTorch-style pseudo-code, modeled after (He et al., 2020), as illustrated in Listing 1.

Listing 1: PyTorch-like pseudo-code of our GCMAE-PT.

```python
# f_q_t, f_q_s: teacher/student q encoders
# f_k: encoders for keys
# d_q_t, d_q_s: teacher/student q decoders
# KQ: dictionary as a queue of keys
# m: momentum, tmp: temperature

f_k_t.params = f_q_t.params # Initialize the encoder for keys
for g_h, A_h, g_l, A_l in loader:
    q_h, k_h = GraphAug(x_h)
    q_l, k_l = GraphAug(x_l)
    aug_h = Mix(q_h, k_h) #Num of samples is 2N
    aug_l = Mix(q_l, k_l)
    mask_h = EmbeddingInsert(aug_h) # Generate masked samples
    mask_l = EmbeddingInsert(aug_l)

    m_e_h = f_q_t.forward(mask_h)
    m_e_l = f_q_s.forward(mask_l)
    m_rec_h, A_rec_h = d_q_t.forward(ReMask(m_e_h)) # Remask the embedding
    m_rec_l, A_rec_l = d_q_s.forward(ReMask(m_e_l))

    loss_rec_t = Reconstruct_Loss(m_rec_h, g_h, A_rec_h, A_h)
    loss_rec_s = Reconstruct_Loss(m_rec_l, g_l, A_rec_l, A_l)
    loss_rec = loss_rec_t + loss_rec_s

    q_m_rec_h, k_m_rec_h = Split(m_rec_h) # Split out q and k
    q_m_rec_l, k_m_rec_l = Split(m_rec_l)

    q_h_ex = f_q_t.proj(Mix(q_h, q_m_rec_h)) # Construct contrastive views
    q_l_ex = f_q_s.proj(Mix(q_l, q_m_rec_l))
    k_hl_ex = KQ(f_k.proj(Mix(k_h, k_m_rec_h)) + f_k.proj(Mix(k_l,
        k_m_rec_l))
    k_hl_ex = k_hl_ex.detach()

    pos_t = bmm(q_h_ex.view(N,1,C), k_hl_ex.view(N,C,4))
    neg_t = mm(q_h_ex.view(N,C),KQ.view(C,K))
    pos_s = bmm(q_l_ex.view(N,1,C), k_hl_ex.view(N,C,4))
    neg_s = mm(q_l_ex.view(N,C),KQ.view(C,K))

    logits_t = cat([pos_t, neg_t], dim=1)
    logits_s = cat([pos_s, neg_s], dim=1)

    labels = zeros(N) # positives are 0-th
    loss_cl_t = CE_Loss(logits_t/tmp, labels)
    loss_cl_s = CE_Loss(logits_s/tmp, labels)
    loss_cl = loss_cl_t + loss_cl_s

    loss = loss_rec + loss_cl
    loss.backward()
    update(f_q_t.params, f_q_s.params)

    f_k.params = m*f_k_t.params+(1-m)*f_q_t.params+(1-m)*f_q_s.params

    enqueue(KQ, k_hl_ex)
    dequeue(KQ)
```

**The Collection of Notations**

| | |
|---|---|
| $\mathcal{G}^h$ | A set of high-density EEG graphs |
| $\mathcal{V}^h$ | A set of high-density EEG nodes (The Complete/HD set) |
| $\mathcal{A}^h$ | A set of high-density EEG adjacency matrices |
| $v_i^h$ | The i-th node in the high-density EEG graph |
| $\mathcal{G}^l$ | A set of low-density EEG graphs |
| $\mathcal{V}^l$ | A set of low-density EEG nodes (The Remaining/LD set) |
| $\mathcal{A}^l$ | A set of low-density EEG adjacency matrices |
| $v_i^l$ | The i-th node in the low-density EEG graph |
| $\mathcal{V}^d$ | The Deleted/Dropped/Removed set |
| $v^d$ | The Deleted/Dropped/Removed node |
| $\mathcal{M}_Q^T$ | The teacher encoder for query graphs |
| $\mathcal{M}_Q^S$ | The student encoder for query graphs |
| $\mathcal{D}_Q^T$ | The teacher decoder for query graphs |
| $\mathcal{D}_Q^S$ | The student decoder for query graphs |
| $\bar{\mathcal{M}}_Q^T$ | The pre-trained teacher encoder for query graphs |
| $\bar{\mathcal{M}}_Q^S$ | The pre-trained student encoder for query graphs |
| $\mathcal{M}_K^T$ | The encoder for teacher key graphs |
| $\mathcal{M}_K^S$ | The encoder for student key graphs |
| $\mathcal{Q}^h$ | A set of high-density EEG query graphs |
| $q^h$ | A high-density EEG query sample |
| $\mathcal{K}^h$ | A set of high-density EEG key graphs |
| $k^h$ | A high-density EEG key sample |
| $\mathcal{Q}^l$ | A set of low-density EEG query graphs |
| $q^l$ | A low-density EEG query sample |
| $\mathcal{K}^l$ | A set of low-density EEG key graphs |
| $k^l$ | A low-density EEG key sample |
| $\widehat{\mathcal{G}}^h$ | A set of augmented high-density EEG graphs |
| $\widehat{\mathcal{G}}^l$ | A set of augmented low-density EEG graphs |
| $\mathcal{G}_m^h$ | A set of high-density masking graphs |
| $\mathcal{G}_m^l$ | A set of low-density masking graphs |
| $e_i$ | A inserted learnable node embedding |
| $\tilde{\mathcal{X}}$ | The reconstructed node feature |
| $\tilde{\mathcal{X}}^{tr}$ | The transpose of reconstructed node feature |
| $\tilde{\mathcal{A}}$ | The reconstructed adjacency matrix |
| $\tilde{\mathcal{G}}^h$ | The reconstructed high-density EEG graph |
| $\tilde{\mathcal{G}}^l$ | The reconstructed low-density EEG graph |
| $\tilde{\mathcal{Q}}^h$ | The reconstructed high-density query graph |
| $\tilde{\mathcal{Q}}^l$ | The reconstructed low-density query graph |

| | |
|---|---|
| $\tilde{\mathcal{K}}^h$ | The reconstructed high-density key graph |
| $\tilde{\mathcal{K}}^l$ | The reconstructed low-density key graph |
| $\mathcal{Q}_{ex}^h$ | The extended high-density EEG query graphs |
| $\mathcal{Q}_{ex}^l$ | The extended low-density EEG query graphs |
| $\mathcal{K}_{ex}^h$ | The extended high-density EEG key graphs |
| $\mathcal{K}_{ex}^l$ | The extended low-density EEG key graphs |
| $\mathcal{K}_{ex}^{hl}$ | A key samples pool |
| $\mathcal{K}_{ex}^{hl+}$ | The positive key samples in the queue |
| $\mathcal{K}_{ex}^{hl-}$ | The negative key samples in the queue |
| $\mathcal{L}_{Rec}^T$ | The reconstruction loss for teacher encoder pre-training |
| $\mathcal{L}_{Rec}^S$ | The reconstruction loss for student encoder pre-training |
| $\mathcal{L}_{Rec}^{Joint}$ | The reconstructed loss for joint teacher and student encoders pre-training |
| $\mathcal{L}_{cl}^T$ | The contrastive loss for teacher encoder pre-training |
| $\mathcal{L}_{cl}^S$ | The contrastive loss for student encoder pre-training |
| $\mathcal{L}_{cl}^{Joint}$ | The contrastive loss for joint teacher and student encoders pre-training |
| $\mathcal{Z}_{ij}$ | The similarity between node i and node j |
| $\mathcal{P}_{ij}^+$ | The total positive node pairs |
| $\mathcal{P}_{ij}^-$ | The total negative node pairs |

