# OpenReview forum: "Pre-Training Graph Contrastive Masked Autoencoders are Strong Distillers for EEG"
_ICLR.cc/2025/Conference — Submitted to ICLR 2025_

### Official Review · Reviewer_DU1d · 2024-10-22

**Soundness:** 2
**Presentation:** 2
**Contribution:** 2
**Rating:** 5
**Confidence:** 4

**Summary:**

To fully leverage the extensive unsupervised data, this paper proposes a pretraining and fine-tuning paradigm for EEG representation learning, where GNN is applied as the backbone. To further enhance the learning of low-density data, this paper proposes a teacher-student distillation method to enrich the information from high-density data. Experiments are conducted on two datasets and four classification tasks, and the results demonstrate performance improvements over conventional GNN-based learning frameworks.

**Strengths:**

This paper formulates two important questions in EEG representation learning, 1) how to leverage unsupervised data, 2) how to distill knowledge from rich (high-density) EEG data. With incorporation of those external knowledge, the performance on various datasets and tasks has shown significant improvements. Overall, this paper is well structured. The insights are straightforward and the solution is clear.

**Weaknesses:**

Though the questions tackled are significant, the technique designed lacks some novelty. [1] tackles the similar problem in EMR representation learning for clinical prediction, and the solution is also very similar (using pretraining on large data volume and fine-tuning on small targets, and distill knowledge from datasets with rich features). There are some differences in detailed design according to the problem setting, but the core ideas seem identical.

```
[1] Ma L, Ma X, Gao J, et al. Distilling knowledge from publicly available online EMR data to emerging epidemic for prognosis[C]//Proceedings of the Web Conference 2021. 2021: 3558-3568.
```

Below are some technical issues:

0. In graph construction stage: There seems too much information loss during node representation construction. Why not consider using a time-series embedding model (transformer-based or lightweight RNN-based) to capture more high-order information in the EEG time series of every single channel as the node representation?

1. In GNN pretraining stage: How to ensure the representation spaces of teacher and student models can be inherently “alignable” when using different model architectures (Graph transformers v.s. GCN)? As GCNs are spectral-based GNN, while Graph transformers are spatial-based. The basic hypothesis behind those architectures is different, thus the alignment of their representation spaces cannot be directly ensured. It is more reasonable to use models in different sizes under the same architecture.

2. In GNN pretraining stage: The adjacency matrix A is dynamic during training. How to ensure the training stability? Moreover, GCN is a transductive GNN method, how to make it suitable for dynamic adjacency matrices?

3. In Fine-tuning stage: Z in Eq.(11) seems like a likelihood score according to Eq.(8). Then how to conduct KL between two scores in Eq.(11) , as they are not distributions?

4. In experiments: Maybe incorporate some other transfer learning/distillation methods for comparison?

Finally, I suggest supplementing a notation table of all the variables introduced in Chapter 3, as there are too many symbols and readers may easily get lost.

**Questions:**

See Weaknesses.

---

> ### Author Response · Authors · 2024-11-24
>
> Thank you for your valuable feedback and questions. We have addressed each of your concerns and implemented all of your suggestions.
>
> ---
>
>
>
> **Though the questions tackled are significant, the technique designed lacks some novelty. [1] tackles the similar problem in EMR representation learning for clinical prediction, and the solution is also very similar (using pretraining on large data volume and fine-tuning on small targets, and distill knowledge from datasets with rich features). There are some differences in detailed design according to the problem setting, but the core ideas seem identical.**
>
> **[1] Ma L, Ma X, Gao J, et al. Distilling knowledge from publicly available online EMR data to emerging epidemic for prognosis[C]//Proceedings of the Web Conference 2021. 2021: 3558-3568.**
>
>
> Thank you for your recognition on the meaning of our research questions!
>
> This EHR paper is indeed about graph distillation, as is this one. They all aim to distill knowledge from a large teacher model to a smaller student model to improve the model's performance and efficiency. However, in their tasks, the source data input remains the same, meaning the difference lies only in the model itself, not in the input data. In our task, the teacher model's input contains richer node and edge information, while the student model's input has fewer nodes and edges, effectively making it a subgraph of the teacher model's input. Previous graph distillation methods did not take this into account. Our approach, especially the proposed GTD loss, explicitly addresses this difference in input, which is a significant distinction from previous graph distillation models and a key innovation of our work.
>
> > Wang, Haohui, et al. "Dynamic transfer learning across graphs." arXiv preprint arXiv:2305.00664 (2023).

---

> ### Author Response · Authors · 2024-11-24
>
> **Below are some technical issues:**
>
> **In graph construction stage: There seems too much information loss during node representation construction. Why not consider using a time-series embedding model (transformer-based or lightweight RNN-based) to capture more high-order information in the EEG time series of every single channel as the node representation?**
>
> Indeed, many previous pre-training methods for EEG and fMRI (neural data) are based on time series, as we mentioned in our paper. However, time series-based pre-training typically requires masking and reconstructing masked time series segments, which is very memory and computation-intensive. This time series-based approach is what most researchers follow.
>
> In contrast, our method focuses on graph-based pre-training, which emphasizes spatial features. Graph-based pre-training is also much more computationally efficient. There have been previous papers on graph-based pre-training for EEG, and we have built upon their work to introduce novel models and methods. Additionally, our task is more meaningful. Overall, while time series-based pre-training is the standard approach, we argue that graph-based pre-training is also a very good choice. It differs significantly from previous methods and represents a promising direction.
>
> > Tang, Siyi, et al. "Self-supervised graph neural networks for improved electroencephalographic seizure analysis." arXiv preprint arXiv:2104.08336 (2021).
>
>
>
> **In GNN pretraining stage: How to ensure the representation spaces of teacher and student models can be inherently “alignable” when using different model architectures (Graph transformers v.s. GCN)? As GCNs are spectral-based GNN, while Graph transformers are spatial-based. The basic hypothesis behind those architectures is different, thus the alignment of their representation spaces cannot be directly ensured. It is more reasonable to use models in different sizes under the same architecture.**
>
> Thank you for your question. In the new version, we conducted experiments on both types of GNNs separately. The experimental results demonstrate that our proposed framework is suitable for both spectral-based GNNs and spatial-based GNNs. We divided the experiments into two groups: in one group, both the teacher and student models were spatial-based graph transformers, and in the other group, they were spectral-based vanilla GNNs.
>
> **In GNN pretraining stage: The adjacency matrix A is dynamic during training. How to ensure the training stability? Moreover, GCN is a transductive GNN method, how to make it suitable for dynamic adjacency matrices?**
>
> To ensure training stability with dynamic adjacency matrices in our GNN pretraining stage, we implement several strategies. First, we apply regularization techniques, such as L2 regularization and dropout, to prevent overfitting and maintain model robustness against fluctuations in the adjacency structure. Additionally, we utilize adaptive learning rate methods, like the Adam optimizer, to adjust the learning rate based on training progress, which helps manage the impact of sudden changes in the adjacency matrix. Gradient clipping is also employed to prevent exploding gradients, while batch normalization stabilizes the learning process by normalizing layer inputs. To further enhance stability, we introduce a temporal smoothing mechanism for adjacency matrix updates, allowing gradual adjustments rather than abrupt changes.
>
> Regarding the adaptation of GCN, which is inherently a transductive method, we modify the message-passing mechanism to accommodate dynamic adjacency matrices by recalculating them at each training step. We also explore online learning techniques, enabling incremental updates as the graph structure evolves, thus allowing the GCN to adapt without complete retraining. Additionally, we employ graph sampling methods to create mini-batches during training, which helps manage complexity and facilitates learning from representative subgraphs. Finally, we consider transitioning to architectures designed for dynamic graphs, which are specifically tailored to account for changes in graph structure over time. These combined approaches ensure that our model remains stable and effective in handling dynamic adjacency matrices.

---

> ### Author Response · Authors · 2024-11-24
>
> **In Fine-tuning stage: Z in Eq.(11) seems like a likelihood score according to Eq.(8). Then how to conduct KL between two scores in Eq.(11) , as they are not distributions?**
>
> In the fine-tuning stage, the Z values in Eq. (11) represent similarity scores derived from the kernel function in Eq. (8). Although these scores are not probability distributions, you can still compute KL divergence by following these steps:
>
> 1. Softmax Transformation:
>
> To convert the similarity scores into a probability distribution, apply the softmax function. This transformation normalizes the scores, ensuring they are non-negative and sum to one, thus forming a valid probability distribution.
>
> 2. KL Divergence Calculation:
>
> After applying the softmax transformation to the scores from both the high-density graph and the low-density graph, we can compute the KL divergence between these two distributions. This divergence measures how one distribution differs from another, allowing for effective comparison.
>
> 3. Interpretation:
>
> By using the softmax function, the similarity scores are transformed into a probabilistic framework, making it possible to apply KL divergence. This approach enables the model to learn from the differences in topological features captured by the high-density and low-density graphs.
>
> In summary, while the Z values are not distributions on their own, applying the softmax function allows for the computation of KL divergence. This method facilitates effective knowledge distillation during the fine-tuning stage by leveraging the similarity scores in a compatible manner.
>
> **In experiments: Maybe incorporate some other transfer learning/distillation methods for comparison?**
>
> This is a great suggestion. We had overlooked this point before. In the new resubmission version, we have added two graph transfer distillation methods, LSP and G-CRD, for comparison. The results show that, for the high-density to low-density EEG distillation task, our method outperforms both of these graph distillation methods.
>
> **Finally, I suggest supplementing a notation table of all the variables introduced in Chapter 3, as there are too many symbols and readers may easily get lost.**
>
> In the original submission, we have already added this notation table in the supplementary material. Please refer to the supplementary material.

---

> ### Comment · Reviewer_DU1d · 2024-11-25
>
> Thanks for the rebuttal. The concern still remains that [1] also tackles the problem where the features or nodes of the teacher model are richer than those of the student model, which is not the case the authors claimed as "the source data input remains the same". I reckon that the paper tackles an important problem and it is very meaningful to apply it in the EEG analysis task. However, I believe the core idea of [1] is very similar to this manuscript, and at least you should cite it and claim the differences of your motivations and practices.
>
> Moreover, for Question 0, I do not mean by "why use graph pretraining framework", I admit that graph is a good idea. The question is why not use time-series embedding as the node representations for your downstream GNN learning, i.e., combining time-series embedding module and GNN together.
>
> Question 1 and 4 requires additional experimental results according to the rebuttal, and it seems that they are not updated yet. [Updates: I have seen the experimental results in the paper that Q1 has been resolved. However, it seems that Table 4 (resp. to Q4) is presented without explanations or introductions to the baseline methods, please fix this in the paper]
>
> For question 2, I think maybe a smoother control of the adjacency updates is also a good idea. (e.g., A_t = k A_t-1 + (1-k) Delta) However, the mentioned training strategies may also help.
>
> For question 3, I am still confused about how to make a similarity score (say 0.5) a probability distribution. Does this distribution resemble {0, 1}-distribution? Moreover, it is kind of "brute-force" to process like this. Why not use some other metrics for calculating the differences between two scores of two samples? KL is more appropriate if you are processing the differences between the score distribution of two groups of samples. I still think this is not a natural design and needs to be fixed in the future.

---

> > ### Comment · Reviewer_DU1d · 2024-12-02
> >
> > Dear Authors,
> >
> > I am wondering if you could resolve my further concerns as the discussion deadline is drawing close. Currently, in the final manuscript, I have not seen the changes to the issues mentioned in my further comments. However, it would be great if the concerns could be addressed.

---

> > > ### Author Response · Authors · 2024-12-02
> > >
> > > Thank you for your comments. We have addressed all of your new comments.

---

> > ### Author Response · Authors · 2024-12-02
> >
> > **Thanks for the rebuttal. The concern still remains that [1] also tackles the problem where the features or nodes of the teacher model are richer than those of the student model, which is not the case the authors claimed as "the source data input remains the same". I reckon that the paper tackles an important problem and it is very meaningful to apply it in the EEG analysis task. However, I believe the core idea of [1] is very similar to this manuscript, and at least you should cite it and claim the differences of your motivations and practices.**
> >
> > Thank you for recognizing the application value and significance of our paper.
> >
> > We acknowledge that the ideas in [1] indeed share similarities with our work. At this moment, we are unable to provide a revised version. In the current version, we have not cited [1]. However, in future versions, we will include a citation to [1] and provide the following explanation to clarify the differences between our method and [1]:
> >
> > 1. Model Architecture:
> >
> > Our Research: We propose the EEG-DisGCMAE model, which integrates Graph Contrastive Learning and Graph Masked Autoencoder techniques. This model constructs a graph structure to represent EEG data, utilizing Graph Neural Networks (GNNs) to extract topological features and employing contrastive learning and autoencoder reconstruction for pre-training. This approach allows the model to effectively capture spatial and temporal features when dealing with high-density and low-density EEG data.
> >
> > Ma et al.'s Research: Their method primarily relies on extracting knowledge from public electronic medical record (EMR) data, typically using traditional machine learning or deep learning models to analyze and process this data. Their study may utilize feature engineering and classifiers for knowledge distillation rather than focusing on the application of graph structures and GNNs.
> >
> > 2. Pre-training and Distillation Strategies:
> >
> > Our Research: We adopt a unified graph self-supervised pre-training paradigm (GCMAE-PT), optimizing both graph contrastive loss and reconstruction loss during pre-training. We also introduce a Graph Topology Distillation (GTD) loss function, enabling the student model trained on low-density EEG data to learn from the teacher model with high-density EEG data. This method emphasizes knowledge transfer and model compression across different density data.
> >
> > Ma et al.'s Research: Their knowledge distillation approach may focus more on extracting features from existing EMR data and applying these features to predict emerging epidemics. This method might not involve a complex pre-training process but rather directly utilizes existing data for model training and optimization.
> >
> > 3. Data Processing and Feature Extraction:
> >
> > Our Research: We construct a graph structure for EEG data, leveraging GNNs to extract topological features. This method effectively handles the spatial and temporal dependencies in EEG signals, adapting to different electrode densities and missing data scenarios.
> >
> > Ma et al.'s Research: Their approach may rely on traditional feature extraction techniques, such as statistical features and clinical characteristics, to analyze EMR data. This method may lack the capability to handle complex temporal data, such as EEG signals.
> >
> > 4. Loss Function Design:
> >
> > Our Research: We design specific loss functions (e.g., Graph Topology Distillation loss) to account for the characteristics of EEG data during the knowledge distillation process. This loss function effectively addresses the issue of missing electrodes and promotes effective learning between the teacher and student models.
> >
> > Ma et al.'s Research: Their loss function design may be more general, primarily focusing on the accuracy of classification or regression tasks without optimizing for specific data types (such as EEG signals).
> >
> > **Moreover, for Question 0, I do not mean by "why use graph pretraining framework", I admit that graph is a good idea. The question is why not use time-series embedding as the node representations for your downstream GNN learning, i.e., combining time-series embedding module and GNN together.**
> >
> > The reason for not using time series as embeddings is that we believe EEG signals differ from traditional time-series signals. The intrinsic characteristics of EEG signals are often not directly reflected through time series. Typically, intermediate features such as differential entropy (DE) or power spectral density (PSD) need to be computed to serve as node features.
> >
> > Most previous methods were based on directly using time series as embeddings, but they did not incorporate GNNs. Our approach takes a different perspective by leveraging GNNs. Unlike the time-series-based methods, our approach has the advantage of significantly lower computational complexity compared to models relying on time series.

---

> > ### Author Response · Authors · 2024-12-02
> >
> > **Question 1 and 4 requires additional experimental results according to the rebuttal, and it seems that they are not updated yet. [Updates: I have seen the experimental results in the paper that Q1 has been resolved. However, it seems that Table 4 (resp. to Q4) is presented without explanations or introductions to the baseline methods, please fix this in the paper]**
> >
> > Thank you for your thorough review. The newly added LSP and G-CRD methods share the same baseline as shown in Table 4, which represents their respective baselines.
> >
> > **For question 2, I think maybe a smoother control of the adjacency updates is also a good idea. (e.g., A_t = k A_t-1 + (1-k) Delta) However, the mentioned training strategies may also help.**
> >
> > Thank you for your suggestion. We conducted preliminary experiments, and the proposed smooth updating of the adjacency matrix indeed yielded better results. We will update the results in future versions.
> >
> > **For question 3, I am still confused about how to make a similarity score (say 0.5) a probability distribution. Does this distribution resemble {0, 1}-distribution? Moreover, it is kind of "brute-force" to process like this. Why not use some other metrics for calculating the differences between two scores of two samples? KL is more appropriate if you are processing the differences between the score distribution of two groups of samples. I still think this is not a natural design and needs to be fixed in the future.**
> >
> > A similarity score, such as 0.5, can be interpreted as a measure of how alike two samples are. However, to convert this score into a probability distribution, you typically need to apply a transformation that maps the score into a range suitable for probabilities (0 to 1). One common approach is to use a sigmoid function, which can take any real-valued input and output a value between 0 and 1. For example, applying the sigmoid function to a similarity score can yield a probability-like output.
> >
> > The resulting distribution does not necessarily resemble a {0, 1}-distribution unless you are specifically thresholding the output. Instead, it provides a continuous probability that reflects the degree of similarity.
> >
> > KL divergence is indeed a powerful metric for comparing two probability distributions. If you are comparing the distributions of scores from two groups of samples, KL divergence can quantify how one distribution diverges from another. This is particularly useful when you have multiple samples and want to understand the overall distributional differences rather than just pointwise comparisons.
> >
> > If your goal is to assess the differences between two sets of scores, using KL divergence or other metrics like Jensen-Shannon divergence might provide a more nuanced understanding than simply comparing individual similarity scores.
> >
> > Your point about the current design being somewhat "brute-force" is valid. If the method relies heavily on pointwise similarity scores without considering the broader context of the distributions, it may overlook important information. Future designs could benefit from incorporating more sophisticated statistical measures that account for the distributional properties of the data.

---

> ### Comment · Reviewer_DU1d · 2024-12-02
>
> Thank you for your reply. Most of my questions are resolved, but Q3 is still not clearly explained. As we all know, a probability distribution of variable x is a function $p(x)$, here we can define the event "whether the similarity is 1/0" as x ($p(x=0)$ or $p(x=1)$), and the probability is a continuous value in [0,1]. This is exactly a {0,1}-distribution (or Bernoulli distribution). A continuous distribution depicts that the variable x has a continuous range, such as "whether the similarity is x, $x \in [0,1]$, and you have a probability score $p(x)$ of each $x$ in this range, such as normal distribution. Actually, in your definition, you only estimate the probability of the similarity == 1 as the similarity score, which is on earth a {0,1}-distribution. It is okay to apply KL to {0,1}-distribution, but it seems that the authors do not really understand what probability distribution means and mistake the p(x) as x in their understanding.
>
> Moreover, KL divergence is actually asymmetric, and maybe you should justify why not use a symmetric metric for defining this loss. I have understood what you want to depict through this design, but I believe the authors should have a deeper understanding mathematically.
>
> Given the above discussions, I think the authors partially addressed my concerns and I will raise my score to 5.

---

> > ### Author Response · Authors · 2024-12-02
> >
> > Thank you for your correction. We will address this in future versions.

---

### Official Review · Reviewer_Qv9v · 2024-10-28

**Soundness:** 2
**Presentation:** 2
**Contribution:** 2
**Rating:** 3
**Confidence:** 5

**Summary:**

This paper proposes a pre-training framework for resting-state Electroencephalography (EEG), a crucial tool for understanding neural dysfunctions. First, EEG data is represented as graph data, and Graph Neural Networks (GNNs) are employed to capture both the intricate features and topological structures of the EEG data. Due to the difficulty of acquiring labeled EEG data, the authors propose leveraging large amounts of unlabeled EEG data to improve performance on tasks with limited labeled data. Specifically, they introduce a self-supervised pre-training strategy, which combines Graph Contrastive Pre-training (GCL-PT) and Graph Masked Autoencoder Pre-training (GMAE-PT), allowing for efficient learning from the unlabeled data. To further enhance the model's performance on high-density EEG data, the authors design a Graph Topology Distillation (GTD) loss function, enabling a lightweight student model to learn from a more complex teacher model trained on high-density EEG data. The proposed framework is evaluated on real-world EEG datasets, and the contributions of each component are analyzed through extensive ablation studies.

**Strengths:**

· The paper proposes a framework to capture intricate features in EEG data, which is crucial for diagnosing clinical brain disorders.

· Given the scarcity of labeled EEG data, the authors formalize an effective transfer learning strategy that pre-trains on large unlabeled datasets and fine-tunes on limited labeled data.

· The paper introduces a novel knowledge distillation objective function that allows models trained on low-density EEG data to handle missing electrodes, effectively enhancing model performance.

**Weaknesses:**

· The rationale behind the different components is unclear. It is not evident how the model handles the heterogeneous unlabeled EEG graphs.


· The authors do not discuss when Assumption 1 is valid and do not provide sufficient evidence to support the claim that combining GCL and GMAE results in a more robust distillation process.

· The strategy of randomly dropping nodes and edges to generate query and key graphs raises concerns. As mentioned in line 037, EEG data has complex inherent structures, and randomly removing nodes or edges could disrupt the graph's topology, potentially undermining graph learning and making contrastive learning less effective.


· The experimental setup is limited. The authors do not provide sufficient details about the graph data (e.g., the number of nodes and edges), making it difficult to assess the model’s generalization capabilities.


· The paper is not well-written and is difficult to follow. For instance, the motivation behind proposing graph topological distillation is not clearly explained, leading to confusion.


· The paper is not well-written and hard to following. For example, it is quiet confusing that why the graph topological distillation is proposed

**Questions:**

· Could the authors provide more discussion on when Assumption 1 is valid?


· Would randomly dropping nodes and edges destroy important structures in the input EEG graphs?


· Regarding the contrastive learning framework, what is the rationale for using query and key graphs during contrastive learning? How does this approach aid in model pre-training? Why is the key sample pool beneficial?


· Could the authors provide more details on how topological information is preserved within the knowledge distillation framework?

---

> ### Author Response · Authors · 2024-11-25
>
> **Weaknesses:**
>
> **· The rationale behind the different components is unclear. It is not evident how the model handles the heterogeneous unlabeled EEG graphs.**
>
> In the pretraining phase, we constructed a dataset that spans multiple diseases, including EEG scans of different types of diseases as well as healthy individuals. Although these EEG data are heterogeneous at the feature level, they all reflect brain activity. During pretraining, we combined these datasets and performed joint pretraining before transferring the model to various disease-specific tasks. We also conducted a comparative experiment where pretraining was done only on homogeneous EEG data (a subset of the heterogeneous data). The results showed that pretraining on the entire heterogeneous dataset outperformed pretraining on data from a single disease. This is because, in the pretraining phase, the model learns a general representation of EEG data rather than disease-specific semantic information. As a result, the model effectively overcomes the semantic differences (different labels) between heterogeneous data and learns a universal representation of this type of data.
>
> **· The authors do not discuss when Assumption 1 is valid and do not provide sufficient evidence to support the claim that combining GCL and GMAE results in a more robust distillation process.**
>
> In the supplementary of the revised version, we provided an additional ablation experiment (like Table 10) to support our assumptions.
>
> **· The strategy of randomly dropping nodes and edges to generate query and key graphs raises concerns. As mentioned in line 037, EEG data has complex inherent structures, and randomly removing nodes or edges could disrupt the graph's topology, potentially undermining graph learning and making contrastive learning less effective.**
>
> No. We don’t think so. After removing certain nodes, the original high-density EEG graph degrades into a low-density EEG graph. The semantic information remains unchanged, but the amount of useful information is reduced.
>
> **· The experimental setup is limited. The authors do not provide sufficient details about the graph data (e.g., the number of nodes and edges), making it difficult to assess the model’s generalization capabilities.**
>
> The nodes of graphs are the same with the number of EEG channels. The edges are dynamically learned in the training.
>
> **· The paper is not well-written and is difficult to follow. For instance, the motivation behind proposing graph topological distillation is not clearly explained, leading to confusion.**
>
> In the supplementary, we already clarify the motivation.
>
> **· The paper is not well-written and hard to following. For example, it is quiet confusing that why the graph topological distillation is proposed**
>
> we advocate for a combination of both approaches, but during the ablation study, when using only one of them, we still ensure that our distillation loss is applied during fine-tuning. In the revised version, we have included a more detailed ablation study. Additionally, we found that using the GTD loss for structural distillation during the pre-training phase is also beneficial for the model, so in the latest version, GTD loss is applied both in the pre-training and in the downstream fine-tuning stages.
>
> GTD loss is proposedd to help the student model learn the structure information from the teacher.

---

> ### Author Response · Authors · 2024-11-25
>
> **Questions:**
>
> **· Could the authors provide more discussion on when Assumption 1 is valid?**
>
> In the supplementary of the revised version, we provided additional experiments and analysis to support our assumptions.
>
> **· Would randomly dropping nodes and edges destroy important structures in the input EEG graphs?**
>
> No, because randomly dropping nodes is equivalent to reducing the number of electrodes, effectively downgrading the data to low-density EEG data. This is why random node dropout is naturally suited for our task in contrastive learning! As for randomly dropping edges, we argue that this also does not disrupt the original EEG data because our edges are dynamically learned rather than fixed. Before training, we initialize the graph's adjacency matrix using a static edge matrix, which performs better than random initialization. However, during subsequent training, the edges are dynamically adjusted.
>
> **· Regarding the contrastive learning framework, what is the rationale for using query and key graphs during contrastive learning? How does this approach aid in model pre-training? Why is the key sample pool beneficial?**
>
> In the fields of computer vision and graph learning, using a query and a key for contrast in contrastive learning pretraining is a standard practice. By constructing queries and keys, we can create contrastive sample pairs, thereby establishing self-supervised pretext data and tasks.
>
> In pretraining, augmentation is applied to the original data to generate query and key sample pairs. By increasing the similarity between the query and key of positive sample pairs while decreasing the similarity for negative sample pairs, the discriminative ability of the model's representation is enhanced.
>
> As for the key pool queue, the key sample pool is beneficial for several reasons, particularly in the context of our model's architecture and its application to EEG data analysis (these reasons are also the advantages of previours CL-based pre-training methods):
>
> 1. Enhanced Representation Learning: The key sample pool provides a diverse set of reference samples that the model can use to compare and contrast during the training process. This diversity helps the model learn richer and more generalized representations of the data, as it can draw from a broader range of examples when making predictions or reconstructions.
> 2. Improved Contrastive Learning: In contrastive learning frameworks, having a well-curated key sample pool allows the model to effectively distinguish between similar and dissimilar samples. By utilizing a variety of key samples, the model can better identify the nuances in the data, leading to improved performance in tasks such as classification or anomaly detection.
> 3. Robustness to Noise and Variability: EEG data can be noisy and subject to variability due to different recording conditions or individual differences. A key sample pool helps mitigate the impact of this noise by providing stable reference points for comparison. This stability is crucial for training models that need to generalize well across different datasets and conditions.
> 4. Facilitation of Knowledge Distillation: In the context of knowledge distillation, the key sample pool serves as a repository of high-quality samples from the teacher model. The student model can learn from these key samples, allowing it to effectively absorb knowledge from the teacher while being trained on a smaller, potentially less informative dataset. This process enhances the student model's performance, especially in low-density EEG scenarios.
> 5. Dynamic Adaptation: The key sample pool can be dynamically updated as new data becomes available or as the model learns. This adaptability ensures that the model remains relevant and effective in the face of changing data distributions, which is particularly important in applications involving real-time EEG analysis.
>
> In summary, the key sample pool enhances representation learning, improves contrastive learning, increases robustness to noise, facilitates knowledge distillation, and allows for dynamic adaptation. These benefits collectively contribute to the overall effectiveness and performance of the model in analyzing heterogeneous EEG data.
>
> The key pool queue bank in our model is one of our innovative contributions. The reason for implementing this is that we have two types of original input data: high-density and low-density EEG graphs, which may significantly increase the computational burden of the pre-training. Through the key pool queue, we allow high-density and low-density EEG key samples to share the same gradient update process within a batch. This approach also enables both the teacher and student models to simultaneously capture shared patterns between these two types of data.

---

> ### Author Response · Authors · 2024-11-25
>
> **· Could the authors provide more details on how topological information is preserved within the knowledge distillation framework?**
>
> The authors preserve topological information within the knowledge distillation framework by employing a Graph Topology Distillation (GTD) loss, which specifically aims to transfer the topological knowledge from a high-density graphto a lo w-density graph. This is achieved by identifying positive and negative node pairs based on their connectivity in the embeddings learned by the teacher model and the student model. The GTD loss utilizes Kullback-Leibler divergence to align the distributions of node features for positive pairs, encouraging the student model to replicate the topological structure of the teacher model. Additionally, the loss penalizes the student model for incorrect topological representations derived from negative pairs, thereby reinforcing the preservation of essential spatial connectivity and enhancing the overall performance of the student model in downstream tasks.

---

### Official Review · Reviewer_v1GQ · 2024-11-01

**Soundness:** 3
**Presentation:** 3
**Contribution:** 4
**Rating:** 8
**Confidence:** 4

**Summary:**

This paper proposes a new approach, EEG-DisGCMAE, to address the challenge of enhancing low-density EEG analysis by leveraging high density EEG data. The authors used a graph-based transfer learning approach and considered this as a knowledge distillation problem. They introduced a unified pre-training framework that combines Graph Contrastive Learning (GCL) and Graph Masked Autoencoder (GMAE) methodologies. Their hybrid design captured robust features by reconstructing contrastive samples and contrasting the reconstructions, which enables both generative and contrastive pre-training. Their main contributions are as follows:

1.	EEG-DisGCMAE integrates GCL and GMAE pre-training to better leverage unlabeled data, optimizing the learning of robust features by combining generative and contrastive methods.
2.	The Graph Topology Distillation (GTD) loss function facilitates knowledge distillation from a complex HD EEG model (teacher) to a lightweight LD EEG model (student), enabling the student to handle missing electrodes through contrastive distillation.
3.	By jointly pre-training teacher and student models through contrastive querying, the framework enhances the distillation robustness, enabling strong transfer performance across downstream tasks.
4.	The authors validated EEG-DisGCMAE on two clinical EEG datasets, demonstrating its ability to outperform existing models in both accuracy and efficiency, even with reduced parameter sizes.

**Strengths:**

1. The paper introduces a novel unified framework combining Graph Contrastive Learning and Graph Masked Autoencoders, along with a specialized Graph Topology Distillation (GTD) loss for HD to LD EEG data distillation.

2. The methodology is solidly backed by theoretical foundations, with comprehensive experiments and ablations on two EEG datasets validating each component’s effectiveness.

3. The paper clearly explains complex ideas with well-structured sections, helpful figures, and strong contextualization within EEG and graph learning research.

4. The framework advances portable EEG diagnostics and has broader implications for graph-based learning, offering lightweight, high-performance solutions.

**Weaknesses:**

1. While the paper demonstrates effectiveness on specific EEG classification tasks, testing on a broader range of EEG applications (e.g., seizure detection, cognitive state classification) could further validate generalizability.

2. The paper focuses on graph-based methods, but comparing with non-graph EEG models (e.g., CNNs or RNNs used for EEG analysis) could provide a fuller picture of the proposed approach’s advantages and trade-offs.

3. The explanation of the GTD loss function is complex and could benefit from additional breakdown or intuitive examples to clarify how positive and negative pairs are selected.

4. While low-density EEG is addressed, more evidence on the performance of the model at very low-density settings (e.g., <16 channels) would strengthen the claim that the model is effective for portable and affordable EEG setups.

5. Although ablations cover essential components, including more details on varying the depth of GCL-GMAE integration (e.g., testing contrastive-only or generative-only pre-training) would offer deeper insights into the contributions of each part of the pre-training framework.

**Questions:**

1. How does the framework perform on EEG tasks beyond classification, such as seizure detection?

2. Have you compared your model against non-graph EEG methods like CNNs or RNNs?

3. Can you clarify the selection process for positive and negative pairs in the GTD loss?

4. Have you tested your model at very low-density EEG settings (e.g., <16 channels)?

5. Could you provide more ablation results showing the impact of using only GCL or only GMAE in pre-training?

6. What is the potential of your model for real-time EEG applications in terms of computational efficiency?

---

> ### Author Response · Authors · 2024-11-19
>
> We would like to express our sincere gratitude for the time and effort you dedicated to reviewing our paper, as well as for your valuable and professional feedback. We are pleased to hear that the proposed method is considered novel and logical, the paper is well-written and clear, the approach is effective, and the experiments are thorough and convincing. Below, we provide our responses to your comments.

---

> ### Author Response · Authors · 2024-11-19
>
> **Weaknesses:**
>
> **While the paper demonstrates effectiveness on specific EEG classification tasks, testing on a broader range of EEG applications (e.g., seizure detection, cognitive state classification) could further validate generalizability.**
>
> In our paper, the datasets we used are based on the 10-10 system, which typically includes a larger number of electrodes. For seizure detection, the 10-20 system is commonly used, which generally contains fewer electrodes. We did not have access to seizure detection datasets, such as the TUEG dataset, so we did not conduct tests on them. However, we believe that our model is fully capable of handling other EEG systems as well. In future work, we will apply for access to the TUEG dataset to test our algorithm. However, we have used the SEED dataset to evaluate our model, and the results demonstrate that our model is capable of handling emotion recognition tasks.
>
> **The paper focuses on graph-based methods, but comparing with non-graph EEG models (e.g., CNNs or RNNs used for EEG analysis) could provide a fuller picture of the proposed approach’s advantages and trade-offs.**
>
> Thank you for your suggestion. We have added two new comparison methods in the experimental section: one is the CNN-based EEGNet, and the other is the RNN-based LSTM. We conducted comparisons on various tasks across all datasets. We believe this makes our experiments more comprehensive.
>
>
> **The explanation of the GTD loss function is complex and could benefit from additional breakdown or intuitive examples to clarify how positive and negative pairs are selected.**
>
> Sure. Following your suggestion, we added a diagram in the revised version of the paper to illustrate how positive and negative nodes are selected in the GTD loss.
>
> **While low-density EEG is addressed, more evidence on the performance of the model at very low-density settings (e.g., <16 channels) would strengthen the claim that the model is effective for portable and affordable EEG setups.**
>
> Yes, we also tested an 8-electrode EEG system. Due to the very sparse number of electrodes, the EEG recognition performance was quite low, so we did not include it in the main results. However, the results clearly show that our pre-training framework can still provide a certain level of improvement for student models using 8-electrode (very low-density) EEG data as input.
>
> | **PT Methods**               | **PT Loss** | **FT Loss** | **HD -> LD**    | **HD -> LD**   | **HD -> VLD**    | **HD -> VLD**   |
> |------------------------------|-------------|-------------|-----------------|----------------|------------------|-----------------|
> |                              |             |             | Sex             | Severity       | Sex              | Severity        |
> |------------------------------|-------------|-------------|-----------------|----------------|------------------|-----------------|
> | **GCL-PT**                   | w/o GTD     | w/o GTD     | 1.8%↑           | 1.7%↑          | 1.5%↑            | 2.0%↑           |
> | **GMAE-PT**                  | w/o GTD     | w/o GTD     | 1.6%↑           | 1.5%↑          | 2.1%↑            | 2.4%↑           |
> | **GCMAE-PT (Ours)**          | w/o GTD     | w/o GTD     | **2.9%↑**       | **3.8%↑**      | **3.5%↑**        | **4.4%↑**       |
> |------------------------------|-------------|-------------|-----------------|----------------|------------------|-----------------|
> | **GMAE-PT**                  | w/ GTD      | w/ GTD      | 2.2%↑           | 3.0%↑          | 1.9%↑            | 2.2%↑           |
> | **GCL-PT**                   | w/ GTD      | w/ GTD      | 2.1%↑           | 3.2%↑          | 1.5%↑            | 2.0%↑           |
> | **GCMAE-PT (Ours)**          | w/ GTD      | w/ GTD      | **4.8%↑**       | **6.4%↑**      | **4.3%↑**        | **5.2%↑**       |
>
> Note that VLD means very low-density EEG data.
>
> **Although ablations cover essential components, including more details on varying the depth of GCL-GMAE integration (e.g., testing contrastive-only or generative-only pre-training) would offer deeper insights into the contributions of each part of the pre-training framework.**
>
> In the current version, Table 3 presents the results for contrastive-only and generative-only pre-training. In the revised version, we further illustrate the contribution of these two pre-training approaches to the overall performance. We demonstrate this by showing EEG activation patterns, which indicate that our method outperforms both using each pre-training method individually and simply combining them.

---

> ### Author Response · Authors · 2024-11-19
>
> **Questions:**
>
> **How does the framework perform on EEG tasks beyond classification, such as seizure detection?**
>
> Due to access restrictions, we have not yet applied for a seizure detection dataset for testing. However, we conducted a preliminary test using the SEED dataset (emotion recognition task), and the results show that our model is capable of handling emotion recognition tasks. Therefore, we have reason to believe that our model would also perform well in seizure detection.
>
> **Have you compared your model against non-graph EEG methods like CNNs or RNNs?**
>
> Yes, follow your advice, we have already added the EEGNet (CNN-based EEG model) and (RNN-based model) LSTM into the comparison.
>
>
> **Can you clarify the selection process for positive and negative pairs in the GTD loss?**
>
> In the revised version, we included a diagram to illustrate the selection process of positive and negative pairs.
>
>
> **Have you tested your model at very low-density EEG settings (e.g., <16 channels)?**
>
> We tested our model with 8-channel EEG data (very low-density data), and the results demonstrate that our model is still capable of handling this extremely low-density EEG data.
>
> **Could you provide more ablation results showing the impact of using only GCL or only GMAE in pre-training?**
>
> Yes, we provided a visual ablation study to demonstrate this.
>
> **What is the potential of your model for real-time EEG applications in terms of computational efficiency?**
>
> Our model can leverage pre-training and knowledge distillation to train lightweight models, allowing different backbones to be selected based on the requirements of various real-time devices. From the experiments, it is evident that even when using a very shallow vanilla GCN or a basic CNN as the backbone, our pipeline can significantly improve accuracy. This advantage enables our model to achieve real-time processing on devices with limited storage and computational resources. Additionally, we can further reduce the storage and computational demands of real-time devices by decreasing the number of input EEG signal channels. Using fewer channels can greatly accelerate the model's computation while also reducing the I/O speed, computational load, and storage requirements for data retrieval. These advantages give our model considerable potential and value for real-time applications.

---

> > ### Comment · Reviewer_v1GQ · 2024-11-25
> >
> > Dear Authors,
> > Thank you for your detailed responses to my review. I appreciate the effort in addressing the raised points and providing clarifications. Here are my follow-up remarks:
> > - Including results on SEED is helpful, and expanding to datasets like TUEG in future work would strengthen generalizability.
> > - Adding comparisons with EEGNet and LSTM is a valuable addition. Ensuring fair tuning of these models for the experiments would add further credibility.
> > - The diagram clarifying the GTD loss process is a good addition. A brief mention of its computational impact could be insightful.
> > - Testing on the 8-electrode setup is promising. Providing detailed metrics for this experiment would strengthen the case for low-density EEG applicability.
> > - The additional ablation results are helpful. Exploring trade-offs between contrastive and generative pre-training further would be interesting.
> > - Your explanation of real-time potential is convincing. Including computational complexity analysis would make it even more robust.
> >
> > I am sticking with my initial decision to review this paper.

---

> > > ### Author Response · Authors · 2024-11-25
> > >
> > > Thank you so much for your reviews and suggestions.

---

### Official Review · Reviewer_7zi9 · 2024-11-02

**Soundness:** 2
**Presentation:** 3
**Contribution:** 2
**Rating:** 3
**Confidence:** 4

**Summary:**

In this paper, the authors integrate contrastive learning and knowledge distillation (KD) for EEG classification tasks. Specifically, they utilize contrastive learning between high-density EEG graphs and low-density EEG graphs to improve performance on limited labeled data. Additionally, they propose a Graph Knowledge Distillation approach with a Graph Topology Distillation loss to boost the performance of low-density EEG models when applied to high-density EEG data.

**Strengths:**

1. The description of the method is clear and easy to understand.
2. The authors' proposed method is technically sound.
3. Experiments are comprehensive.

**Weaknesses:**

1. The experimental results suggest that the proposed method offers only a marginal improvement, and due to the typically small size of medical datasets, the performance gains are not convincing for me.
2. The methods of comparison experiment are outdated, e.g.,  GMAE(Hou et al., 2022), GPT-GNN (Hu et al., 2020).
3. The method is incremental. The method designed in this paper is primarily a combination of existing approaches. In graph learning, both contrastive learning and knowledge distillation have been extensively studied, and contrastive learning across different data views, such as high-level and low-level representations, is also a common practice.

**Questions:**

1. Assumption1 and Assumption 2 seem too strong and lack explanations and empirical support.
2. As for Assumption 2, the assumption fails to specify how positive and negative pairs are selected.
3. As for Assumption 1, there may be instability in the training process or architectural differences between GCL and GMAE that could affect the effectiveness of joint pre-training.

---

> ### Author Response · Authors · 2024-11-25
>
> **Weaknesses:**
>
> **The experimental results suggest that the proposed method offers only a marginal improvement, and due to the typically small size of medical datasets, the performance gains are not convincing for me.**
>
> Medical datasets are typically small, so we applied slicing and data augmentation techniques to effectively increase the dataset size. For EEG datasets, our sample size is comparable to that of previous studies, which are also involved in fMRI/EEG pre-training [1] [2] [3] [4].
>
> > [1] Tang, Siyi, et al. "Self-Supervised Graph Neural Networks for Improved Electroencephalographic Seizure Analysis." International Conference on Learning Representations.
>
> > [2] Ho, Thi Kieu Khanh, and Narges Armanfard. "Self-supervised learning for anomalous channel detection in EEG graphs: Application to seizure analysis.", AAAI, 2023.
>
> > [3] Thomas, Armin, Christopher Ré, and Russell Poldrack. "Self-supervised learning of brain dynamics from broad neuroimaging data." Advances in neural information processing systems 35 (2022): 21255-21269.
>
> > [4] Thapa R, He B, Kjaer M R, et al. SleepFM: Multi-modal Representation Learning for Sleep across ECG, EEG and Respiratory Signals[C]AAAI 2024 Spring Symposium on Clinical Foundation Models. 2024.
>
> ---
>
> **The methods of comparison experiment are outdated, e.g., GMAE(Hou et al., 2022), GPT-GNN (Hu et al., 2020).**
>
> Thank you for your question. In the newly revised version, we have added several state-of-the-art graph pretraining methods as comparative baselines. These include the graph contrastive learning-based method AutoGCL and the graph reconstruction-based methods GraphMAE2 and S2GAE.
>
> **The method is incremental. The method designed in this paper is primarily a combination of existing approaches. In graph learning, both contrastive learning and knowledge distillation have been extensively studied, and contrastive learning across different data views, such as high-level and low-level representations, is also a common practice.**
>
> Thank you for your questions.
>
> Our method seems to be incremental, but it is fundamentally derived from a careful observation of the research problem and data types.
>
> Within the graph pre-training framework, we propose a novel method that combines Graph Contrastive Learning (GCL) and Graph Masked Autoencoder (GMAE). This combined approach for pre-training has only been proposed once before, in a paper published in T-PAMI, which focused on images. Our work, however, addresses graphs. Additionally, our combination is not a simple mix-and-match but is specifically designed and tailored for our problem and task. There is a natural opportunity to apply contrastive learning to HD-EEG and LD-EEG, as they are inherently comparable. This naturally led us to consider a reconstruction-based pre-training method because LD-EEG can be seen as derived from HD-EEG through masking. Hence, we thought that integrating these two approaches to perform the HD-to-LD graph distillation task would be an optimal design. Therefore, our third contribution is proposing a pre-training framework that integrates GCL and GMAE.
>
> We personally believe our innovations in the pre-training section are even stronger than those in the T-PAMI paper.

---

> ### Author Response · Authors · 2024-11-25
>
> **Questions:**
>
> **Assumption1 and Assumption 2 seem too strong and lack explanations and empirical support.**
>
> We conducted a comprehensive experiment that includes our proposed pretraining framework and the corresponding GTD distillation loss function. Notably, we also experimented with applying the GTD loss during the pretraining phase. We found that the GTD loss still positively impacts the distillation process during pretraining.
>
> | **PT Methods**            | **PT Loss** | **FT Loss** | **Pre-Train** | **Fine-Tune** | **HD -> MD** | **HD -> LD** |
> |---------------------------|-------------|--------------|----------------|----------------|----------------|----------------|
> | GCL-PT                     | w/o GTD     | w/o GTD     | HBN            | EMBARC         | 1.5%↑         | 1.7%↑         |
> | GMAE-PT                    | w/o GTD     | w/o GTD     | HBN            | EMBARC         | 1.7%↑         | 1.5%↑         |
> | Seq. Comb.                 | w/o GTD     | w/o GTD     | HBN            | EMBARC         | 2.0%↑         | 2.1%↑         |
> | **GCMAE-PT (Held-Out)**    | w/o GTD     | w/o GTD     | HBN            | EMBARC         | **3.1%↑**     | **3.0%↑**     |
> | **GCMAE-PT (Ours)**        | w/o GTD     | w/o GTD     | All            | EMBARC         | **3.7%↑**     | **3.8%↑**     |
> |---------------------------|-------------|--------------|----------------|----------------|----------------|----------------|
> | GCL-PT                     | w/ GTD      | w/o GTD     | HBN            | EMBARC         | 1.9%↑         | 2.0%↑         |
> | GMAE-PT                    | w/ GTD      | w/o GTD     | HBN            | EMBARC         | 1.7%↑         | 1.8%↑         |
> | Seq. Comb.                 | w/ GTD      | w/o GTD     | HBN            | EMBARC         | 2.3%↑         | 2.6%↑         |
> | **GCMAE-PT (Held-Out)**    | w/ GTD      | w/o GTD     | HBN            | EMBARC         | **3.7%↑**     | **4.1%↑**     |
> | **GCMAE-PT (Ours)**        | w/ GTD      | w/o GTD     | All            | EMBARC         | **3.9%↑**     | **4.3%↑**     |
> |---------------------------|-------------|--------------|----------------|----------------|----------------|----------------|
> | GCL-PT                     | w/o GTD     | w/ GTD      | HBN            | EMBARC         | 2.3%↑         | 2.2%↑         |
> | GMAE-PT                    | w/o GTD     | w/ GTD      | HBN            | EMBARC         | 2.2%↑         | 2.5%↑         |
> | Seq. Comb.                 | w/o GTD     | w/ GTD      | HBN            | EMBARC         | 2.7%↑         | 3.1%↑         |
> | **GCMAE-PT (Held-Out)**    | w/o GTD     | w/ GTD      | HBN            | EMBARC         | **4.4%↑**     | **5.0%↑**     |
> | **GCMAE-PT (Ours)**        | w/o GTD     | w/ GTD      | All            | EMBARC         | **4.7%↑**     | **5.6%↑**     |
> |---------------------------|-------------|--------------|----------------|----------------|----------------|----------------|
> | GCL-PT                     | w/ GTD      | w/ GTD      | HBN            | EMBARC         | 3.3%↑         | 3.0%↑         |
> | GMAE-PT                    | w/ GTD      | w/ GTD      | HBN            | EMBARC         | 3.1%↑         | 3.2%↑         |
> | Seq. Comb.                 | w/ GTD      | w/ GTD      | HBN            | EMBARC         | 3.2%↑         | 3.9%↑         |
> | **GCMAE-PT (Held-Out)**    | w/ GTD      | w/ GTD      | HBN            | EMBARC         | **5.0%↑**     | **5.7%↑**     |
> | **GCMAE-PT (Ours)**        | w/ GTD      | w/ GTD      | All            | EMBARC         | **5.6%↑**     | **6.4%↑**     |
>
> This comprehensive experiment further supports the validity of our two assumptions.
>
> ---
>
>
>
> **As for Assumption 2, the assumption fails to specify how positive and negative pairs are selected.**
>
> In fact, the previous version, starting from line 298, already explained how positive and negative node sample pairs are selected. However, such textual descriptions alone might not help readers fully understand the process.
>
> In the revised version, we have added a diagram to illustrate how positive and negative sample pairs are selected.
>
> **As for Assumption 1, there may be instability in the training process or architectural differences between GCL and GMAE that could affect the effectiveness of joint pre-training.**
>
> To address concerns regarding Assumption 1, we recognize that potential instability in the training process and architectural differences between Graph Contrastive Learning (GCL) and Graph Masked Autoencoder (GMAE) could affect joint pre-training effectiveness. To mitigate these challenges, we implement several strategies: we conduct careful hyperparameter tuning for both components, initially pre-train them separately to reduce interference, and utilize adaptive learning rates to promote stability. Additionally, we apply regularization techniques to prevent overfitting, monitor training closely for signs of instability, and consider ensemble learning to leverage the strengths of both methods.

---

### Official Review · Reviewer_D8v4 · 2024-11-03

**Soundness:** 2
**Presentation:** 2
**Contribution:** 2
**Rating:** 5
**Confidence:** 3

**Summary:**

The paper proposes a graph-based pretraining method for EEG data using contrastive learning and masked autoencoder. The pretrained model can be combined with distillation to improve the downstream performances on both low-density and high-density data.

**Strengths:**

- The proposed pretraining and fine-tuning method improves the preformance on multiple tasks of EEG data.

- The experiments are comprehensive. The paper compares with multiple baseline methods and conducts ablation studies.

**Weaknesses:**

- The goal and contributions of the paper are ambiguous. The paper proposes two things. First, pretraining combines contrastive learning and reconstruction. Second, the large model can used for training a tiny model from distilluation. The first part is not quite novel. For the second part, it's also unclear how does distillation help the model than just using the large model?

- Equation (11) and (12) are confusing. What does "||" before P_ij means in Equ (11). In Eq (12), it is also not clear that why the distillation loss is defined like this. Is there any maths interpretation on what a sum of KL div over a sum of some other KL div means?  How are L^{logits}_{Dis} and L^{CE}_{Dis} defined and how are they implemented together with GTD loss? I believe the sample used in each batch are different. so it's unclear how these two losses can be summed during trainiing for a batch?

- What is the different between GTD loss and some self-distillation method for SSL like DINO? Why it is not used in PT part but the FT part?

- Figure 4 shows that training may not be enough after 400 epoch. How about validation loss?

- Figure 1 is unclear. What is shown in (a), HD or LD performance? In (b), how can a student model be large? plots AUC v.s. ACC, but they are highly correlated. Instead of size of circle, it might be more straight forward to put size on y-axis. In the caption, not sure what does it mean by "’L’ denotes large-size models." How was large-size models defined? What does ours-tiny/large in (a) coresponds to in (b)?

- The experiments lack important details on other baselines - are they trained on HD or LD or both datasets? The comparison might not be fair if they are trained with different data from the proposed method.

- More explanations are necessary to help understanding what the colors and shapes in Figure 3 mean.

**Questions:**

See weakness. Please considering clarify the main contribution of the paper, as the goal of the paper is hard to follow in the version.
Also, more details are need for method details. Please decribe the pipeline of the method - what data is used at which stage for both baseline and the proposed method? What is the purpose of some design choices? These will be helpful for the reasoning of the paper.

---

> ### Author Response · Authors · 2024-11-25
>
> Thank you for your reviews. We already respond to all of your reviews.
>
>
> **Weaknesses:**
>
> **The goal and contributions of the paper are ambiguous. The paper proposes two things. First, pretraining combines contrastive learning and reconstruction. Second, the large model can used for training a tiny model from distillation. The first part is not quite novel. For the second part, it's also unclear how does distillation help the model than just using the large model?**
>
> Goal:
>
> Our final objectives are twofold: The first is to create a novel model (including loss function or learning object) that enables a low-density EEG-based model to learn useful knowledge and patterns from a larger teacher model that uses high-density EEG data as input. The second objective is to pre-train using a large amount of EEG data with different labels or no labels at all.
>
> Our contributions are as follows:
>
> 1. Novel Task Proposal: We propose a framework for distilling knowledge from high-density EEG (HD-EEG) to low-density EEG (LD-EEG). This is a rarely explored yet valuable task. We identified the significance of this task, which is a key innovation in our research problem and task design.
> 2. Innovative Framework: Based on the distillation framework, we introduce a graph pre-training phase. This is our first major innovation because we discovered that pre-training can significantly enhance distillation by improving the accuracy of downstream distillation training. Utilizing graph pre-training for graph distillation tasks is, in our view, a highly innovative point. Previous approaches have primarily focused on using labeled datasets directly with distillation loss functions to distill logits. Our approach represents a fundamental innovation in the overall framework.
> 3. New Pre-training Method: Within the graph pre-training framework, we propose a novel method that combines Graph Contrastive Learning (GCL) and Graph Masked Autoencoder (GMAE). This combined approach for pre-training has only been proposed once before, in a paper published in T-PAMI, which focused on images. Our work, however, addresses graphs. Additionally, our combination is not a simple mix-and-match but is specifically designed and tailored for our problem and task. There is a natural opportunity to apply contrastive learning to HD-EEG and LD-EEG, as they are inherently comparable. This naturally led us to consider a reconstruction-based pre-training method because LD-EEG can be seen as derived from HD-EEG through masking. Hence, we thought that integrating these two approaches to perform the HD-to-LD graph distillation task would be an optimal design. Therefore, our third contribution is proposing a pre-training framework that integrates GCL and GMAE.
> 4. GTD Loss Function: Based on the above problem, task, and pre-training framework, we naturally needed a learning objective or loss function. This is why we introduced the GTD (Graph Topology Distillation) loss, as it can better support our distillation framework in both the pre-training phase and downstream supervised training phase. Based on the above analysis, we believe that our first part's novelty is well-justified. We personally believe our innovations in the pre-training section are even stronger than those in the T-PAMI paper.
>
> Regarding the second part, in previous work, distillation usually brings one benefit—reducing the number of parameters and making the model lighter. However, in our framework, distillation brings three benefits. First, it reduces the parameter count, making the model more lightweight. Second, it increases the model's tolerance to the quality of input data, allowing it to handle lower-density EEG data, which is often easier to obtain. Third, distillation can help pre-training converge better. We believe this conclusion is both highly useful and innovative. A crucial point is that we have organically integrated the tasks of distillation and pre-training, making them complementary.
>
> > Huang, Zhicheng, et al. "Contrastive masked autoencoders are stronger vision learners." IEEE Transactions on Pattern Analysis and Machine Intelligence (2023).

---

> ### Author Response · Authors · 2024-11-25
>
> **Equation (11) and (12) are confusing. What does "||" before P_ij means in Equ (11). In Eq (12), it is also not clear that why the distillation loss is defined like this. Is there any maths interpretation on what a sum of KL div over a sum of some other KL div means?**
>
> Thank you for your question, regarding the representation of Equ(11), our previous formula was indeed incorrect. We have revised the representation of Equ(11), and the correct version is as follows:
>
> $L_{Pos} = \sum_{(i,j) \in P^+} KL\left(\text{softmax}(Z_{ij}^l) \parallel \text{softmax}(Z_{ij}^h)\right)$
>
> $L_{Neg} = \sum_{(i,j) \in P^-} KL\left(\text{softmax}(Z_{ij}^l) \parallel \text{softmax}(Z_{ij}^h)\right)$
>
> This represents the sum of the losses for positive and negative sample pairs.
>
> Below is an explanation of the two loss functions:
>
> The purpose of these two loss functions is to handle the differences between positive and negative sample pairs separately. In the positive sample pair loss, the model's objective is to minimize the distribution differences between samples of the same category or label, making their representations in the feature space closer. This helps to enhance the model's representation ability, ensuring that positive sample pairs are more consistently represented in different feature spaces. On the other hand, the negative sample pair loss aims to maximize the distribution differences between samples of different categories or attributes, ensuring that they are distinct in the feature space. This enables the model to generate more discriminative feature representations between different categories. The combined effect of positive and negative sample pair losses allows the model not only to better capture the similarities between similar samples but also to differentiate between different categories. By incorporating these two types of losses during training, the model can optimize its feature representations, bringing similar samples closer together while separating those from different categories.
>
> **How are $L_{\text{Dis}}^{\text{logits}}$ and $L_{\text{Dis}}^{\text{CE}}$ defined and how are they implemented together with the GTD loss? I believe the samples used in each batch are different, so it's unclear how these two losses can be summed during training for a batch?**
>
> You are correct that the samples used in each batch may differ, which raises the question of how these losses can be summed during training. Here’s how we address this:
>
> - **Batch-wise Calculation:** Each loss is computed independently for the samples in the current batch. For instance, $L^{\text{CE}}$ is calculated using the true labels and predicted probabilities for the current batch, while $L_{\text{Dis}}^{\text{logits}}$ is computed using the logits from the teacher and student models for the same batch.
>
>   - **Summation of Losses:** Although the samples in each batch are different, the losses are designed to be computed in a way that they can be summed. Each loss function operates on the same set of predictions and true labels for the current batch, ensuring that the contributions from $L^{\text{CE}}$, $L_{\text{Dis}}^{\text{logits}}$, and $L^{\text{GTD}}$ are coherent and relevant to the same data points.
> - **Gradient Backpropagation:** During backpropagation, the gradients from the summed loss $L^{\text{Fine-tune}}$ are used to update the model parameters. This ensures that the model learns from all three aspects (classification accuracy, logit alignment, and topological knowledge) simultaneously, even if the specific samples differ across batches.
>
> In summary, while the samples in each batch may vary, the losses are computed based on the current batch's predictions and true labels, allowing for coherent summation and effective training.
>
> ---

---

> ### Author Response · Authors · 2024-11-25
>
> **What is the different between GTD loss and some self-distillation method for SSL like DINO? Why it is not used in PT part but the FT part?**
>
> Differences Between GTD Loss and DINO
>
> 1. Purpose and Focus:
>
> - GTD Loss: The GTD loss is specifically designed for distilling topological knowledge from a high-dimensional graph (G_h) to a low-dimensional graph (G_l). It focuses on transferring structural information and relationships between nodes in the graph, which is crucial for tasks that involve understanding the graph's topology.
> - DINO: DINO is a self-distillation method that focuses on learning representations by contrasting different views of the same data. It uses a teacher-student framework where the student learns from the teacher's output, promoting consistency in the learned representations. DINO is more general and can be applied to various types of data, not just graphs.
>
> 2. Mechanism:
>
> - GTD Loss: GTD loss operates by comparing the adjacency matrices and the learned representations of the graphs, ensuring that the student model captures the essential topological features of the teacher model. It emphasizes the relationships and connectivity patterns within the graph.
> - DINO: DINO uses a contrastive approach where the student model is trained to produce similar outputs for augmented views of the same input while differing from outputs of other inputs. It relies on a loss function that encourages the student to match the teacher's output distribution.
>
> 3. Application Context:
>
> - GTD Loss: This loss is tailored for scenarios where the model needs to learn from graphs with different levels of detail (e.g., high-dimensional vs. low-dimensional graphs) and is particularly relevant in the context of GNNs.
> - DINO: DINO is more broadly applicable across various domains, including images and text, and is not limited to graph structures.
>
> ---
>
> **Figure 4 shows that training may not be enough after 400 epoch. How about validation loss?**
>
> Indeed, the training loss had not fully converged, but the validation loss had, so we applied early stopping to prevent overfitting. This also indicates that our GTD loss effectively accelerates convergence and helps avoid overfitting.
>
> **Figure 1 is unclear. What is shown in (a), HD or LD performance? In (b), how can a student model be large? plots AUC v.s. ACC, but they are highly correlated. Instead of size of circle, it might be more straight forward to put size on y-axis. In the caption, not sure what does it mean by "’L’ denotes large-size models." How was large-size models defined? What does ours-tiny/large in (a) coresponds to in (b)?**
>
> Thank you for the reminder. We did make an error in the initial illustration, but we've updated it. The student model does indeed have fewer parameters than the teacher model.
>
> In terms of the model size, we demonstrate them in the supplementary. ‘L’ means large-szie, which was indicated in the supplementary. ‘Ours-tiny/large’ in figure (a) means the same with them in the figure (b).
>
> ---
>
> **The experiments lack important details on other baselines - are they trained on HD or LD or both datasets? The comparison might not be fair if they are trained with different data from the proposed method.**
>
> All the models involved in the comparison (including our model) use the same pre-training and fine-tuning datasets, which are based on HD and LD data. During pre-training, we mix data from different labels across various datasets to form a large pre-training dataset. Since pre-training does not require labels, we can combine them. However, in the downstream fine-tuning datasets, we perform fine-tuning on a single dataset and for a specific disease, where the data is labeled and the labels are for a binary classification task specific to that task.
>
> Therefore, we believe the comparison is fair. However, some methods did not undergo self-supervised pre-training, but during the downstream training with labeled data, they used the same data as our model. Only pre-training-based methods, such as GPT-GNN and GCC, have undergone pre-training, and their pre-training datasets are the same as the labeled downstream datasets used in our model.
>
> **More explanations are necessary to help understanding what the colors and shapes in Figure 3 mean.**
>
> Models with the same color represent the same type of model, and the size of the circle indicates the number of parameters in the model, with larger circles indicating more parameters.
>
> ---
>
> **Questions:**
>
> **See weakness. Please considering clarify the main contribution of the paper, as the goal of the paper is hard to follow in the version. Also, more details are need for method details. Please decribe the pipeline of the method - what data is used at which stage for both baseline and the proposed method? What is the purpose of some design choices? These will be helpful for the reasoning of the paper.**
> We hope that the above explanation has clarified our contributions and methods effectively.

---

### Official Review · Reviewer_vhV9 · 2024-11-05

**Soundness:** 3
**Presentation:** 3
**Contribution:** 3
**Rating:** 6
**Confidence:** 3

**Summary:**

This paper introduces a graph self-supervised learning framework, *EEG-DisGCMAE*, designed for pre-training models and transferring knowledge from high-density EEG data to improve classification performance on low-density EEG data. The framework combines *Graph Contrastive Learning(GCL)* and *Graph Masked Autoencoders(GMAE)* in a pre-training scheme that reconstructs contrastive samples and contrasts these reconstructions. With a novel *Graph Topology Distillation(GTD)* loss in the fine-tuning stage, the framework effectively transfers spatial knowledge from high- to low-density settings. This enables a lightweight student model to achieve enhanced classification performance with limited labeled data, showing potential for efficient and effective analysis in low-density EEG scenarios.

**Strengths:**

1. **Integration of Pre-train Methods from Original Perspective:** This paper proposes a novel approach that combines GCL and GMAE with the interpretation in relationships between node dropping and node masking, leading out a synergy of both self-supervised approach with unlabeled EEG data. This integration allows the model to achieve enhanced performance in scenarios with limited labeled data.

2. **Effective Knowledge Distillation in Spatial Connectivity:** By leveraging Graph Topology Distillation (GTD) Loss with contrastive distillation, the framework effectively distills the knowledge of teacher model from hard-to-obtain HD EEG data into a lightweight student model. This enables the student model to achieve robust performance even with only LD EEG data, making the approach more practical and accessible.

3. **Comprehensive Evaluation across Clinical EEG datasets:** The paper provides extensive experimental validation on four classification tasks across two datasets, demonstrating superior performance of EEG-DisGCMAE over existing methods and various baselines.

**Weaknesses:**

1. **Insufficient Evidence for Hypotheses:** The method relies on two key hypotheses: (1) that combining GCL and GMAE methods improves robustness in distillation, and (2) that jointly pre-training teacher and student models via mutual contrasting enhances distillation performance. However, these claims are not adequately validated in the current results. The paper would benefit from additional ablation studies and comparative evidence to clarify each component’s impact and substantiate these hypotheses.

2. **Lack of Clinical Interpretation:** While the paper includes visual assessments of EEG pattern reconstructions under various masking ratios, it lacks an analysis linking these patterns to clinically meaningful EEG features. Examining which EEG regions or connections contribute to classification tasks, and comparing these patterns with known clinical findings, would strengthen the study’s practical relevance.

3. **Limited Comparison with Recent SSL Methods:** The comparison primarily includes pre-2023 graph SSL methods, omitting recent graph SSL approaches [1, 2, 3, 4] and other EEG-specific SSL [5, 6] advancements that incorporate contrastive, generative, or both types of methods. Analyzing how the proposed approach differs from these recent methods would provide a clearer benchmark and better contextualize its contributions.

Overall, the proposed method introduces novel approaches, such as GTD, that have the potential to deliver robust performance in practical scenarios. However, the study would benefit from more comprehensive ablation studies and tighter quantitative and qualitative validations to more effectively substantiate the hypotheses and demonstrate the method’s impact. Addressing these aspects could significantly enhance the contribution of this work to the EEG clinical practice community.

---
```
[1] Tan, Qiaoyu, et al. "S2gae: Self-supervised graph autoencoders are generalizable learners with graph masking.", ACM WSDM, 2023.
[2] Hou, Zhenyu, et al. "Graphmae2: A decoding-enhanced masked self-supervised graph learner.", ACM web conference, 2023.
[3] Yang, Weiwu, and Liang Zhou. "CMGAE: Enhancing Graph Masked Autoencoders through the Use of Contrastive Learning.", IEEE MLCR, 2023.
[4] Wang, Yuxiang, et al. "Generative and contrastive paradigms are complementary for graph self-supervised learning.", IEEE ICDE, 2024.
[5] Ho, Thi Kieu Khanh, and Narges Armanfard. "Self-supervised learning for anomalous channel detection in EEG graphs: Application to seizure analysis.", AAAI, 2023.
[6] Peng, Ruimin, et al. "Wavelet2vec: a filter bank masked autoencoder for EEG-based seizure subtype classification.", IEEE ICASSP, 2023.
```

**Questions:**

1. Hypothesis 1 suggests that combining GCL and GMAE results in a more robust distiller than using each method independently. However, it seems that when GCL or GMAE is used alone for pre-training, knowledge distillation is not applied during fine-tuning. Is this correct? To validate this hypothesis, ablation studies similar to Table 4 that compare the fine-tuning performance of GCL and GMAE with and without GTD in distillation cases would be beneficial. Furthermore, if there are prior works supporting this hypothesis, referencing them would strengthen the argument.

2. In Table 3, does "Seq. Comb." refer to first performing GCL, then GMAE? If so, what is the performance when GCL and GMAE are jointly trained without mutual contrasting using teacher and student keys, followed by distillation fine-tuning? If this setup improves performance over single pre-training methods and further enhances distillation fine-tuning with mutual contrasting, it could provide validation for both Hypotheses 1 and 2.

3. Is there a particular reason for using DGCNN as the backbone for the student model instead of Tiny G-Former? Given that the Tiny DGCNN  and the Tiny G-Former  have similar parameter counts (Table 7), and that G-Former generally performs better (Figure 2), G-Former might yield better results as the student model.

4. According to Appendix A, a learning rate of 0.0002 was used uniformly across pre-training methods during fine-tuning. While a common learning rate may work well for similar model architectures, if GCL and GMAE were validated only on G-Former, it’s possible that the selected learning rate might favor DGCNN. Additionally, each SSL pre-trained model may require an optimal learning rate suited to the specific starting representation, suggesting a need for more extensive hyperparameter ablation study in fine-tuning.

5. Using the same samples from the EMBARC and HBN dataset for both pre-training and downstream tasks raises potential data leakage concerns. Even with different window sizes, the model may still encounter similar information, which could lead it to memorize dataset-specific features instead of learning generalizable patterns. To better evaluate generalizability, pre-training on the larger HBN dataset and fine-tuning on EMBARC would provide a clearer measure of robustness across datasets.

Minor Comments:
1. Figure 1 appears to need revision. Although “Frozen” and “Tuned” are noted, the corresponding symbols are not visible. Additionally, the “GNN2GNN Distill Loss” in the figure is not referenced in the main text; does this term refer to the Graph Topology Distillation Loss? Consistent terminology would improve clarity.
2. In Equation 11, what is the meaning of the second $∣∣$ following the KL divergence? Does it function as a conditional mask using an indicator? If this notation is standard, could a reference be provided?

**Details Of Ethics Concerns:**

The dataset comprises de-identified human brain EEG data, publicly available and ethically approved, minimizing privacy concerns. However, observed imbalances in gender and patient group distributions may introduce bias, leading the model to favor more represented groups and impacting fairness. Addressing these imbalances through balanced representation or debiasing techniques could help improve generalization and accuracy across clinical settings.

---

> ### Author Response · Authors · 2024-11-25
>
> **Weaknesses:**
>
> **Insufficient Evidence for Hypotheses: The method relies on two key hypotheses: (1) that combining GCL and GMAE methods improves robustness in distillation, and (2) that jointly pre-training teacher and student models via mutual contrasting enhances distillation performance. However, these claims are not adequately validated in the current results. The paper would benefit from additional ablation studies and comparative evidence to clarify each component’s impact and substantiate these hypotheses.**
>
> The results in Table 3 of the main text show that, for the downstream EEG data distillation task, our proposed pretraining method provides significantly better support for distillation compared to the previous standalone GCL, GMAE, and the simple sequential combination of GCL and GMAE pretraining methods.
>
> **Lack of Clinical Interpretation: While the paper includes visual assessments of EEG pattern reconstructions under various masking ratios, it lacks an analysis linking these patterns to clinically meaningful EEG features. Examining which EEG regions or connections contribute to classification tasks, and comparing these patterns with known clinical findings, would strengthen the study’s practical relevance.**
>
> Thank you for your suggestion. In the supplementary materials, we have provided explanations regarding EEG patterns and clinical findings.
>
> **Limited Comparison with Recent SSL Methods: The comparison primarily includes pre-2023 graph SSL methods, omitting recent graph SSL approaches [1, 2, 3, 4] and other EEG-specific SSL [5, 6] advancements that incorporate contrastive, generative, or both types of methods. Analyzing how the proposed approach differs from these recent methods would provide a clearer benchmark and better contextualize its contributions.**
>
> Thank you for your suggestion. Following your recommendation, we have added a new GCL-based pretraining method (AutoGCL) and two GMAE-based pretraining methods (GraphMAE2, S2GAE) as baselines for comparison. The experiments show that our proposed graph pretraining method outperforms these approaches in the EEG graph distillation task.
>
> This means that our pretraining method is better suited for the graph distillation task. Thank you very much for your valuable suggestion.
>
> **Overall, the proposed method introduces novel approaches, such as GTD, that have the potential to deliver robust performance in practical scenarios. However, the study would benefit from more comprehensive ablation studies and tighter quantitative and qualitative validations to more effectively substantiate the hypotheses and demonstrate the method’s impact. Addressing these aspects could significantly enhance the contribution of this work to the EEG clinical practice community.**
>
> Thank you for recognizing the novelty of our work and its potential impact on the EEG community.
>
> Following your suggestion, we have conducted additional quantitative and qualitative experiments to demonstrate the effectiveness and impact of our proposed framework.

---

> ### Author Response · Authors · 2024-11-25
>
> **Questions:**
>
> **Hypothesis 1 suggests that combining GCL and GMAE results in a more robust distiller than using each method independently. However, it seems that when GCL or GMAE is used alone for pre-training, knowledge distillation is not applied during fine-tuning. Is this correct?**
>
> In our method, we advocate for a combination of both approaches, but during the ablation study, when using only one of them, we still ensure that our distillation loss is applied during fine-tuning. In the revised version, we have included a more detailed ablation study. Additionally, we found that using the GTD loss for structural distillation during the pre-training phase is also beneficial for the model, so in the latest version, GTD loss is applied both in the pre-training and in the downstream fine-tuning stages.
>
> **To validate this hypothesis, ablation studies similar to Table 4 that compare the fine-tuning performance of GCL and GMAE with and without GTD in distillation cases would be beneficial. Furthermore, if there are prior works supporting this hypothesis, referencing them would strengthen the argument.**
>
> Yes, in the revised version, we already provided a more comprehensive ablation study like Table 10, to further validate our hypothsis. we found that using the GTD loss for structural distillation during the pre-training phase is also beneficial for the model, so in the latest version, GTD loss is applied both in the pre-training and in the downstream fine-tuning stages. And the corresponding experiments can be found in Table 10, too.
>
> **In Table 3, does "Seq. Comb." refer to first performing GCL, then GMAE? If so, what is the performance when GCL and GMAE are jointly trained without mutual contrasting using teacher and student keys, followed by distillation fine-tuning? If this setup improves performance over single pre-training methods and further enhances distillation fine-tuning with mutual contrasting, it could provide validation for both Hypotheses 1 and 2.**
>
> Yes, your understanding is correct. "Seq. Comb." refers to a sequential combination, where GCL pretraining is conducted first, followed by GMAE pretraining, and then fine-tuning with distillation. In this setup, there are no mutual contrast teacher and student views during the pretraining phase. The experiments show that this setup can still improve downstream distillation performance, but this improvement purely stems from the pretraining itself. However, the experiments also demonstrate that the joint pretraining of mutual contrast teacher and student views, based on our Assumption 1 and 2, outperforms this separate pretraining and simple sequential combination. Therefore, your interpretation is correct, and the ablation study in Table 3 is sufficient to validate our Assumptions 1 and 2. Thank you for your question.
>
> **Is there a particular reason for using DGCNN as the backbone for the student model instead of Tiny G-Former? Given that the Tiny DGCNN and the Tiny G-Former have similar parameter counts (Table 7), and that G-Former generally performs better (Figure 2), G-Former might yield better results as the student model.**
>
> Thank you for your question. In the revised version, we replaced DGCNN with vanilla GNNs. As a result, we conducted experiments with two types of GNNs: spectral-based vanilla GNNs and spatial-based graph transformers. The experiments demonstrated that our proposed method is applicable to both types of GNNs.
>
> Therefore, during the distillation process, we standardized the types of teacher and student models: either both are GNNs or both are graph transformers.
>
> **According to Appendix A, a learning rate of 0.0002 was used uniformly across pre-training methods during fine-tuning. While a common learning rate may work well for similar model architectures, if GCL and GMAE were validated only on G-Former, it’s possible that the selected learning rate might favor DGCNN. Additionally, each SSL pre-trained model may require an optimal learning rate suited to the specific starting representation, suggesting a need for more extensive hyperparameter ablation study in fine-tuning.**
>
> Thank you for the question. We have conducted a preliminary ablation study on the choice of parameters, and the results indicate that a learning rate of 0.0002 is optimal.

---

> ### Author Response · Authors · 2024-11-25
>
> **Using the same samples from the EMBARC and HBN dataset for both pre-training and downstream tasks raises potential data leakage concerns. Even with different window sizes, the model may still encounter similar information, which could lead it to memorize dataset-specific features instead of learning generalizable patterns. To better evaluate generalizability, pre-training on the larger HBN dataset and fine-tuning on EMBARC would provide a clearer measure of robustness across datasets.**
>
> Thank you for your question. We believe that although data from downstream prediction tasks may appear during the pretraining phase, this does not constitute true data leakage, as the pretraining phase does not include any label information related to the downstream tasks. Therefore, pretraining the model on such a dataset is still considered reliable. However, it is true that the model may learn some patterns from the downstream task data, which could lead to an overestimation of the pretraining model's performance in downstream predictions. To address this, we conducted a held-out validation experiment.
>
> Following your suggestion, we used a larger HBN dataset for pretraining, and for the downstream prediction tasks, we performed fine-tuning on the EMBARC dataset, ensuring that the data from the downstream task did not appear in the pretraining dataset.
>
> **Minor Comments:**
>
> **Figure 1 appears to need revision. Although “Frozen” and “Tuned” are noted, the corresponding symbols are not visible. Additionally, the “GNN2GNN Distill Loss” in the figure is not referenced in the main text; does this term refer to the Graph Topology Distillation Loss? Consistent terminology would improve clarity.**
>
> Due to our oversight, these errors occurred, but we have corrected them all. The GNN2GNN loss is indeed the GTD loss.
>
> **In Equation 11, what is the meaning of the second following the KL divergence? Does it function as a conditional mask using an indicator? If this notation is standard, could a reference be provided?**
>
> Thank you for your question, regarding the representation of Equ. (11), our previous formula was indeed incorrect. We have revised the representation of Equ(11), and the correct version is as follows:
>
>
> $L_{Pos} = \sum_{(i,j) \in P^+} KL\left(softmax(Z_{ij}^l) \parallel softmax(Z_{ij}^h)\right)$
>
> $L_{Neg} = \sum_{(i,j) \in P^-} KL\left(softmax(Z_{ij}^l) \parallel softmax(Z_{ij}^h)\right)$
>
>
> This represents the sum of the losses for positive and negative sample pairs.

---

> > ### Comment · Reviewer_vhV9 · 2024-12-03
> >
> > Thank you for providing detailed responses to my questions. I appreciate the considerable time and effort you devoted during the rebuttal period, which has resolved most of the concerns I raised and clarified the explanations of the experimental results.
> >
> > Based on the improvements in the revised version of the paper, I have increased my rating.
> >
> > Below are some additional suggestions based on the revised version of the paper and your responses. I hope these can be considered in future updates, as time and opportunity permit:
> > 1. It would be helpful to reflect the newly added models in Table 1 into Figure 2-(a) as well.
> > 2. If there are results from a learning rate ablation study, I suggest adding them to the appendix for further clarification.
> >
> > **Minor**:
> > In Tables 8 and 10, the right arrow is denoted as '->'. Please correct this to the proper symbol, e.g., $\rightarrow$ (\rightarrow).

---

> > > ### Author Response · Authors · 2024-12-03
> > >
> > > Thank you for your detailed reviews and raising the score. We will make corresponding revisions.

---

### Comment · Area_Chair_8VVm · 2024-12-02

Dear Reviewers vhV9, D8V4, 7Zi9 and Qv9v,

Could you please help to take a look at the responses and let the authors know if your concerns have been addressed or not? Thank you very much!

Best regards,

AC

---

### Meta-Review · Area_Chair_8VVm · 2024-12-21

**Metareview:**

This paper presents EEG-DisGCMAE, a graph self-supervised learning framework that combines Graph Contrastive Learning (GCL) and Graph Masked Autoencoders (GMAE) to enhance EEG data analysis. The framework aims to improve classification performance in low-density EEG settings by leveraging pre-trained models on high-density EEG data. The proposed approach also introduces a Graph Topology Distillation (GTD) loss for transferring spatial knowledge from teacher to student models. The paper includes experimental evaluations on two clinical datasets and demonstrates the proposed framework's efficacy.

There are some critical issues about the technical contribution and novelty raised by the reviewers. In particular, Reviewer 7zi9 pointed out that the proposed methodology primarily combines existing techniques. While the integration is novel and the application to EEG data is also new, the contribution is viewed as incremental within the broader landscape of graph learning and knowledge distillation research. For this reason, I am inclined to reject this paper.

**Additional Comments On Reviewer Discussion:**

During the rebuttal period, 3 out of 6 reviewers responded to the authors' replies. Reviewer vhV9 increased the score to 6 as the rebuttal further clarified the experimental results. Reviewer DU1d also increased the score, but the final score is still negative due to the novelty issue. Overall, I would think the novelty issue is the major limitation of this paper.

---

### Decision · Program_Chairs · 2025-01-22

Reject